# Multiphase separation in postsynaptic density regulated by membrane geometry via interaction valency and volume

**Risa Yamada, Giovanni B Brandani, Shoji Takada***

Department of Biophysics, Graduate School of Science, Kyoto University, Kyoto, Japan

## eLife Assessment

This **important** study provides a conceptual advance in our understanding of how membrane geometry modulates the balance between specific and non-specific molecular interactions, reversing multiphase morphologies in postsynaptic protein assemblies. Using a mesoscale simulation framework grounded in experimental binding affinities, the authors successfully recapitulate key experimental observations in both solution and membrane-associated systems, providing novel mechanistic insight into how spatial constraints regulate postsynaptic condensate organization. The conclusions are supported by **solid** strength of evidence and the findings are of broad significance for both computational and experimental biologists

*For correspondence:
takada@biophys.kyoto-u.ac.jp

**Competing interest:** The authors declare that no competing interests exist.

## Abstract

Biomolecular condensates are found at various cellular locations, nucleus, cytoplasm, and membrane. These condensates often contain multiple components and can separate into multiple phases with various morphologies such as core-shell droplets, implicating functional roles. Demixing and arrangements of condensates are determined by competitive interactions and their locations. Recent studies reported a puzzling multiphase morphology in postsynaptic density components: AMPA receptor, NMDA receptor, PSD-95, and CaMKII. The multiphase morphology appears reversed when transitioning from the solution to the membrane. Using this system as a model, we study the multiphase behavior of condensates in solution (3D) and domain formation on and beneath the membrane (2D) and elucidate molecular mechanisms behind the puzzle. Our simulations reproduce the core-shell structure in 3D in vitro solution, where AMPA-receptor/PSD-95 form the core and NMDA-receptor/CaMKII form the shell, triggered by CaMKII activation. Then, we obtain a reversed morphology on the membrane. This reversal is primarily driven by CaMKII's high valency and large volume. We find that, in solution, CaMKII's non-specific volume interaction dominates, while on the membrane, specific multivalent interactions overcome the excluded volume interaction of CaMKII. The layered structures of receptors and CaMKIIs reduce the excluded volume effects of CaMKII on receptors, making the multivalent interaction dominant. These findings highlight the differences between condensate formation in solution and membrane domain formation, modulated by their layered arrangement.

## Introduction

Biomolecular condensates have gained vast attention in recent biological studies. Among them, biomolecular condensates composed of multiple components often separate into two or even more

phases since the components within the condensate are also immiscible with each other (*Banani et al., 2017*; *Hyman et al., 2014*; *Shin and Brangwynne, 2017*). Condensates with multiple phases can have a variety of morphologies (*Kaur et al., 2021*; *Lafontaine et al., 2021*; *Latham and Zhang, 2022*; *Stoecklin and Kedersha, 2013*), such as core-shell or nested structures in which one phase is completely enveloped by the other, as that formed by the FC, DFC, and GC components of the eukaryotic nucleolus (*Lafontaine et al., 2021*), or structures in which droplets share boundaries and contact each other partially, as in the relationship between P-bodies and stress granules (*Stoecklin and Kedersha, 2013*). These multiphase morphologies have been found to serve as a rather complex partitioning structure within the cell and are closely related to the condensate's function as an organelle, as they can control biochemical reactions through the compartment by regulating or buffering the concentration of biomolecules (*Alberti et al., 2019*; *Banani et al., 2017*; *Conti and Oppikofer, 2022*; *Lyon et al., 2021*).

Apart from macroscopic factors such as temperature, pH, and salt concentration, the morphology of the condensate is innately programmed into its constituent molecules themselves via sequences (*Lin et al., 2018*; *Lin et al., 2017*). Molecular interactions can be divided into attraction and repulsion, which results in associative and segregative phase separations, respectively (*Minton, 2020*; *Zeng and Pappu, 2023*). A major driving force of liquid-liquid phase separation (LLPS) is the attraction between molecules, which can be often achieved by interactions between intrinsically disordered regions or by multivalent domain-wise interactions (*Li et al., 2012*; *Wang et al., 2018*). Condensates caused by associative phase separation are known to move more inward into the phase as the constituent molecules have a higher valency (*Chew et al., 2023*; *Espinosa et al., 2020*; *Zhang et al., 2021*). On the other hand, repulsive excluded volume effects can lead to a segregative phase separation by entropic effects. Even if there are no direct attractive forces between molecules, under crowded environments, excluded volume interactions alone can separate the phase with respect to the volume; the smaller ones move inwards and the larger ones move outwards (*Hu et al., 2024*; *Wei et al., 2017*). The elaborate fractionated structure inside the cell can be realized by the competition of these multivalent attractions and excluded volume-driven entropic effects (*Boeynaems et al., 2019*; *Fare et al., 2021*). However, it is not yet clear how the combination of attractive interactions and entropic effects contributes to the changes of the condensate shape and how it is coupled with local environment.

Postsynaptic density (PSD), a protein condensate that forms beneath the postsynaptic membrane, is an ideal model system to examine the competing attraction and repulsion in multi-phase morphology. First, PSD proteins contain a wide variety of repulsive excluded volume interactions and attractive multivalent interactions. Second, recent studies reported a puzzling multiphase morphology (*Hosokawa et al., 2021*; *Hruska et al., 2022*), which is reversed between the solubilized constructs and the system on and beneath the membrane (as explained below), suggesting subtle competing interactions. PSD is one example of biomolecular condensate that can yield multiple compartment structures (*Chen et al., 2023*; *Zeng et al., 2019*; *Zeng et al., 2018*; *Zhu et al., 2024*). Condensation of PSD proteins is closely related to synaptic function and elicits the formation of a receptor-assembled structure called nanodomain during long-term potentiation (LTP) (*Fukata et al., 2024*; *Liu et al., 2021*; *Nair et al., 2013*). A nanodomain on the postsynaptic membrane enables AMPA receptors (AMPARs) and NMDA receptors (NMDARs) to receive neurotransmitters released from the presynapse more efficiently and to cause firing with a higher probability, compared to the scattered configuration (*Nair et al., 2013*; *Raghavachari and Lisman, 2004*; *Ramsey et al., 2021*). Thus, PSD regulates synaptic plasticity via the receptor nanodomain formation. The mechanism can be regarded as the fundamental process of our memory and learning (*Heine et al., 2008*; *Zeng et al., 2016*). The condensation of PSD is closely associated with various pathologies such as high-pressure neurological syndrome (*Cinar et al., 2020*), further attesting to the broad significance of research into this field.

The formation of such nanodomain structures is triggered by the activation of calcium calmodulin-dependent protein kinase II (CaMKII) (*Hosokawa et al., 2021*). CaMKII is one of the most abundant molecules in PSD and is estimated to contain as many as 5600 monomers per synaptic spine (*Sheng and Hoogenraad, 2007*). Other than its abundance, CaMKII has notable features that contribute to phase separation, namely in its high valency and gigantism. First, CaMKII is a highly multivalent molecule consisting of a dodecamer or a tetradecamer (*Buonarati et al., 2021*; *Myers et al., 2017*). When activated, each of 12 or 14 CaMKII subunits has a strong interaction with the C-terminus of GluN2B (a subunit of NMDAR) near its kinase domain (*Barcomb et al., 2016*; *Bayer et al., 2001*).

This multivalent attraction allows a two-component mixture of GluN2Bc and active CaMKIIα to form a monophasic condensate (*Cai et al., 2023*; *Hosokawa et al., 2021*). Second, the dodecameric CaMKII is a very bulky molecule with a diameter of 15–35 nm; a recent study has reported that extension of CaMKII linker disrupts the formation of the condensate with GluN2Bc and exhibits the homogenous phase (*Cai et al., 2023*). This result suggests that phase separation is disrupted by repulsion due to the volume of CaMKII itself. Taken together, CaMKII has two properties with opposite effects on condensation: a high valency, which promotes LLPS, and gigantism, which represses LLPS.

Recent studies reported that CaMKII activation induces multiphase separation into a phase containing AMPAR and another containing NMDAR, presenting a puzzling multiphase morphology (*Hosokawa et al., 2021*; *Hruska et al., 2022*). In vivo observation of the postsynaptic membrane after LTP elucidated that NMDAR forms nanodomains near the center, whereas the AMPAR domain is distributed at the periphery (*Hosokawa et al., 2021*; *Hruska et al., 2022*). On the other hand, in vitro experiments with soluble counterparts of AMPAR and NMDAR, together with PSD-95 and the active CaMKII, showed a core-shell two-phase droplet with AMPAR and PSD-95 at the core and NMDAR and CaMKII in the shell (*Hosokawa et al., 2021*). Thus, the mixture of four components in solution apparently shows a reversed multiphase morphology to what is observed on the postsynaptic membrane. What interaction causes such an apparently contradictory morphology is currently unknown. Since CaMKII activation induces the multiphase separation, the above-mentioned opposite properties of CaMKII toward LLPS may contribute to the changes in the multiphase morphology.

The purpose of this study is to explore the interplay of molecular interactions for the multiphase condensate morphology and the roles of membrane geometry in the PSD proteins. Using four components of PSD (AMPAR complex, NMDAR, PSD-95, and CaMKII) as a model system, we explored the determinants causing the difference of the multiphase morphology between a soluble construct (3D system) and system on and beneath the membrane (2D system, for brevity). We built a mesoscale model based on available experimental data and performed its simulations with these four nanodomain components of PSD. We first validated the model by reproducing the multiphase morphology observed in experiments in the 3D and 2D systems and found that the CaMKII activation induced the multiphase separation. The interaction network analysis elucidated that, while non-specific interactions dominate in determining the phase behaviors in the 3D system, specific interactions play major roles in the 2D system. Then, we examine the roles of CaMKII in the multiphase behaviors by artificial modulation of the interactions. When we decrease the non-specific volume interaction of the CaMKII in the 3D system, it induces a reversal of the multiphase morphology. Then, in the 2D system, a decrease in the valence of specific interactions of CaMKII induces a reversal of the multiphase morphology. These results suggest that the membrane geometry alleviates the non-specific excluded volume interaction due to the layered organization in the 2D systems, resulting in the dominance of specific multivalent interactions, compared to the 3D system.

## Results
### Mesoscale model for multivalent protein LLPS

We investigate multiphase separation of four components of the PSD upper layer, (1) the glutamate receptor AMPAR (or its soluble counterpart) in complex with its regulatory protein called stargazin (TARP γ–2 subunit, or its soluble C-terminal fragment TARPc), (2) the glutamate receptor NMDAR (or its soluble fragment), (3) the scaffold protein PSD-95, and (4) the giant dodecameric kinase complex CaMKIIα, which may be either active or inactive. As in the previous studies (*Chattaraj et al., 2021*; *Lin et al., 2022*; *Yamada and Takada, 2023*), we represent these four protein complexes with a mesoscale model (*Figure 1*), and perform comparative molecular dynamics (MD) simulations for the mixture of them. The mesoscale model has domain resolution; each globular domain (or segment) is represented by a single spherical particle (*Figure 1ACD*). AMPAR(TARP)$_4$ is represented by five particles; one for AMPAR and four for TARPs. NMDAR is modeled as three particles; one for a tetrameric receptor core and two for the C-terminus of GluN2B (termed GluN2Bc) that contains the PDZ-binding motif (PBM). The PSD-95 model contains six particles, representing the N-terminal segment (which can be palmitoylated), three PDZ domains, the SH3 domain, and the GK domain. CaMKII is represented by 13 particles, 12 kinase domains, and 1 central hub domain, which are mutually connected throughout the simulations.

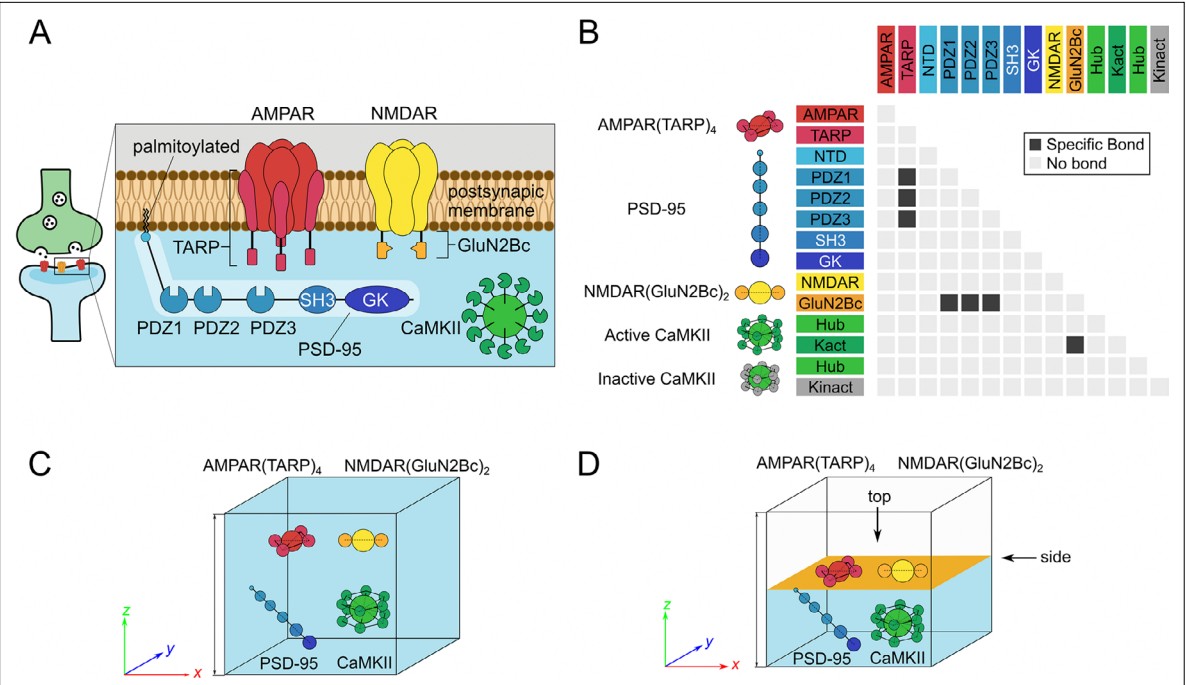

**Figure 1.** Mesoscale model of postsynaptic density proteins. (**A**) Schematics and molecular representation of the four PSD components in this simulation study. AMPAR and NMDAR diffuse on membrane, N-terminal part of PSD-95 is embedded on membrane due to the palmitoylation, and CaMKIIα is present beneath the membrane. (**B**) Specific domain-domain interactions. Every PDZ domain in PSD-95 can interact with TARP and with GluN2Bc. Only active CaMKII can interact with GluN2Bc by its kinase domains. (**C**) The 3D system simulation box. Core spherical particles of AMPAR and NMDAR are substituted by fluorescent proteins DsRed2 and eqFP670, respectively. (**D**) The 2D system setup and simulation box. Particles representing AMPAR, NMDAR, and N-terminal of PSD-95 diffuse on the planar membrane (orange), whereas other particles diffuse in the cytosol beneath the membrane (sky-blue).

The online version of this article includes the following figure supplement(s) for figure 1:

**Figure supplement 1.** Three pairs of molecules with specific interactions between those domains.

One-to-one domain-domain binding is treated as a virtual reaction to rigorously maintain the interaction valency (*Yamada and Takada, 2023*). Interaction strengths between domains are carefully calibrated based on a set of in vitro experiments. Previous in vitro experiments (*Hosokawa et al., 2021*; *Pandey et al., 2025*; *Zeng et al., 2019*) showed that soluble counterparts of these proteins exhibit core-shell (or phase-in-phase) phase separation at an average concentration of 10 μM. The dissociation constant between TARPc and the full length of PSD-95, that between GluN2Bc and the full length of PSD-95, and that between GluN2Bc and the kinase domain of active CaMKII are known to be 2.66 μM (*Vistrup-Parry et al., 2021*), 1.19 μM (*Vistrup-Parry et al., 2021*), and 0.10 μM (*Özden et al., 2022*), respectively (*Figure 1B*, *Figure 1—figure supplement 1*). Note that PBMs in both of TARPc and GluN2Bc interact with the PDZ domains of PSD-95, and thus the binding of two PBMs is mutually exclusive for the binding to the PDZ domain. GluN2Bc interacts with PSD-95 through the PBM, which is the last four residues of GluN2B C-terminus (*Niethammer et al., 1996*), and with CaMKII at 1290–1309 residues (*Bayer et al., 2006*; *Bayer et al., 2001*), allowing GluN2Bc to bind to both PSD-95 and CaMKII simultaneously (see Material and methods for more details).

To validate the simulation method and compare the results with in vitro and in vivo experiments unambiguously, we set up the two simulation systems corresponding to each system. The first system is a mixture containing four soluble protein complexes in solutions, soluble counterparts of AMPAR(TARP)₄ and NMDAR, PSD-95 and CaMKII, dissolved in a cubic simulation box (*Figure 1C*). In this system, based on a previous experiment (*Hosokawa et al., 2021*), AMPAR and TARP are replaced with DsRed2 (PDB ID: 1ZGO, N=936 residues) and a soluble C-terminal part of TARP (TARPc), respectively. NMDAR is also replaced with the soluble fluorescent protein eqFP670 (PDB ID: 4EDS, N=482 residues) and the soluble C-terminal portion of two GluN2B subunits connected to eqFP670 (GluN2Bc). CaMKII can be either in active or inactive state; the active, but not the inactive, CaMKII

binds to GluN2Bc of NMDAR. This system is denoted as the 3D system hereafter. The second simulation system contains the AMPAR(TARP)$_4$ complex and NMDAR both embedded into the membrane plane by force fields, the palmitoylated PSD-95 anchored to the membrane via the palmitoyl group by the same potential, and CaMKII dissolved in the solution under the membrane (*Figure 1D*). Domains of PSD-95 other than palmitoylated N-terminus and CaMKII are not bound to the membrane and diffuse freely below the membrane. This system is denoted as the 2D system for brevity.

## Multiphasic structures of PSD are induced by CaMKII activation

We first performed MD simulations for the mixture of four kinds of PSD proteins starting from random configurations both in the 3D and 2D systems and monitored phase behaviors in comparison with experiments. Based on the previous experiment, the 3D simulation system consists of 135 AMPAR(TARP)$_4$ complexes, 90 NMDARs, 240 PSD-95s, and 60 active CaMKII complexes in a cubic box with 155 nm per side, which corresponds to the concentrations above the estimated critical concentration for the phase separation (*Appendix 1—table 1*). Starting from a random configuration, we observe step-wise growths of the cluster that contains four proteins over tens of milliseconds (*Figure 2A*). Each cluster is nearly spherical, supporting the fluidity of the protein condensates. The stepwise growth is due to the merge of two small droplets into a larger one. In about 20 ms, nearly all AMPAR(TARP)$_4$, NMDAR, and PSD-95s molecules together with ~37 CaMKIIs out of 60 are in one large droplet (*Video 1*). The rest of the space, the dilute phase, contains ~23 CaMKIIs and less than one other protein, on average. Thus, the 3D system exhibits clear phase separation into a high-density droplet and a dilute phase. In general, when the LLPS contains more than one component, its phase transition can be quantified by the solubility product of the components in the dilute phase (*Chattaraj et al., 2021*). For the current four-component PSD system, the product of concentrations of each molecule in the dilute phase is in good agreement with that of the experimental concentrations (*Appendix 1—table 2*).

*Figure 2C* shows the final snapshot of the simulation in the 3D system, which clearly shows a core-shell droplet configuration; a phase containing AMPAR and PSD-95 is at the core, while a phase containing NMDAR, PSD-95, and CaMKII is at the outer layer (*Figure 2D*). These are all consistent with the in vitro experiments (*Hosokawa et al., 2021*; *Pandey et al., 2025*), thus validating our simulation model.

When we substituted the active CaMKIIs into the inactive ones which do not have specific interactions to NMDARs, we observe that AMPAR(TARP)$_4$, NMDAR, and PSD-95, but not CaMKII, coalesce into a large droplet, while nearly all the CaMKII molecules are outside of the droplet forming a dilute phase (*Video 2*, *Figure 2E*). Notably, molecules inside the droplet exhibit a different configuration from the case of the active CaMKII; AMPAR(TARP)$_4$, NMDAR, and PSD-95 are well mixed in the droplet with no demixing (*Figure 2F*). Therefore, the CaMKII activation induces multiphase separation into the AMPAR-containing phase and the NMDAR-containing phase. The selective binding of the active CaMKII to GluN2Bc in NMDAR, but not to AMPAR, induces the multiphase separation. This reproduces the experimental behavior observed in the in vitro solution system reported by Hosokawa and coworkers (*Hosokawa et al., 2021*; *Pandey et al., 2025*).

Next, we performed a simulation mimicking the membrane geometry (2D system) and checked how PSD proteins behave and assemble on and beneath the membrane, using the same interaction parameters and the number of molecules as that in the 3D system. Here, AMPAR(TARP)$_4$, NMDAR(-GluN2Bc)$_2$, and the N-terminal palmitoyl group of PSD-95 are assumed to be embedded to membrane. The length of the cubic box is 500 nm and the receptors diffuse over a 500 × 500 nm membrane plane located at z=0. The number of receptors and the area of membrane are based on the typical areal densities of AMPARs in postsynaptic membranes. In the 2D system, starting from a random configuration, we see modest-sized and dynamic clusters, which repeatedly grow and collapse (*Figure 2B*). As in our previous study that contained only two components (*Yamada and Takada, 2023*), the current 2D system that consists of four components does not show clear separation into macroscopic phases, but transient and dynamic clustering. The two types of clusters, one consisting of AMPAR(TARP)$_4$ and PSD-95, and the other consisting of NMDAR, PSD-95, and CaMKII, form nanoscale domains each of which contains ~50 copies of the two receptors, but does not grow into one macroscopic phase. Both nanodomains with AMPARs and those with NMDARs are unstable; clusters with NMDARs seem to be more stable than those with AMPARs, although both are repeatedly merged and collapsed throughout the simulation time (*Video 3*). Clusters of both receptors shape each distinct area, though

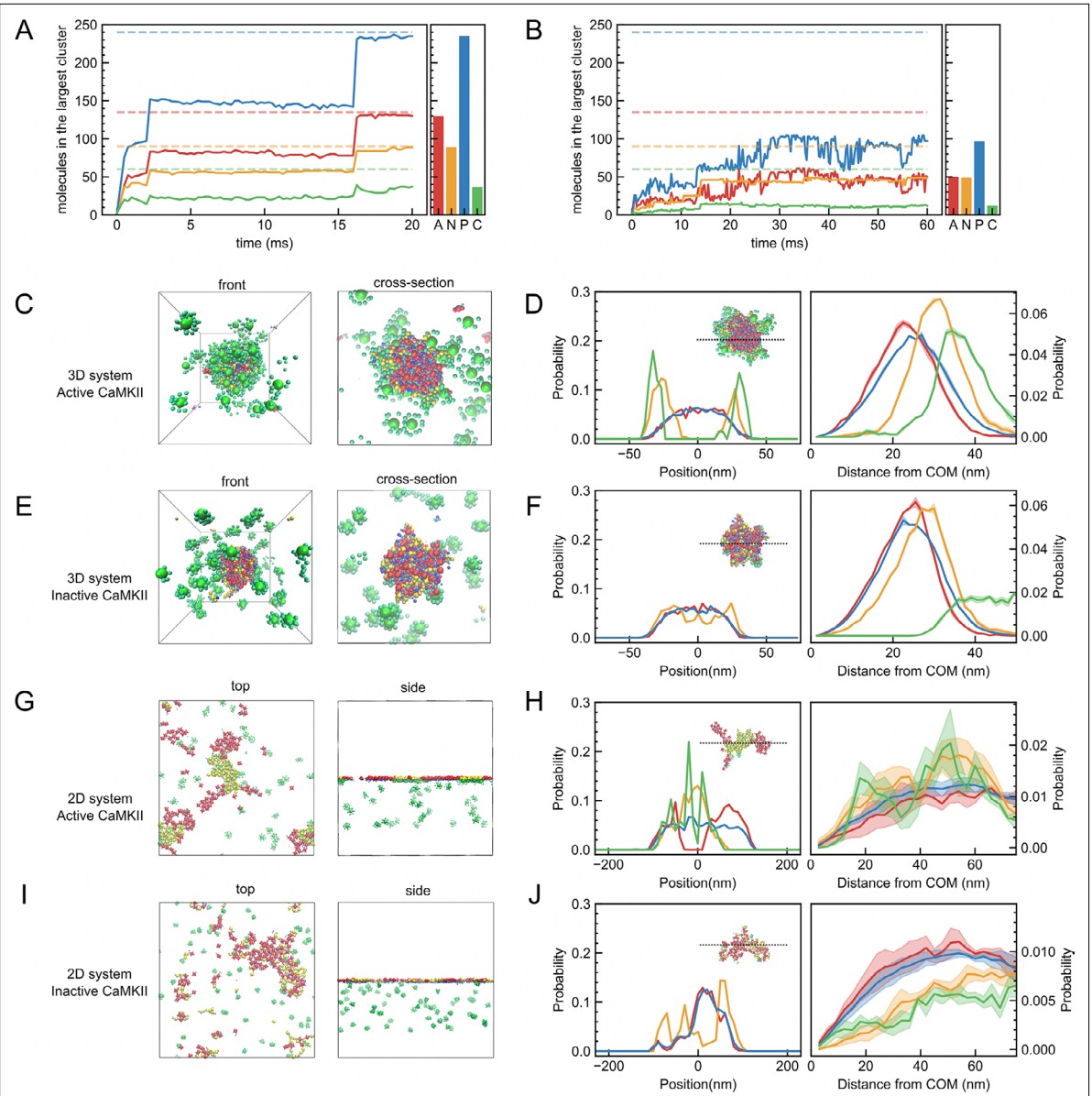

**Figure 2.** Trajectories and morphologies in simulations of four-component PSD systems. (**A, B**) Representative trajectories in the 3D (**A**) and 2D (**B**) systems. A bar chart to the right of the trajectory shows the average molecular compositions of AMPARs (A, red), NMDARs (N, orange), PSD-95s (P, blue), and CaMKIIs (C, green) in the largest cluster in the final 1000 snapshots. Dashed horizontal lines represent the whole number of proteins in the system. (**C, E**) Overview (left) and cross-sectional view (right) final snapshots of simulations in the 3D system with all CaMKIIs in the activated state (**C**) and in the inactive state (**E**). (**D, F**) Density profiles in the slice of the droplet along z axis (left) and the molecular distribution as a function of the distance from the center of mass of the cluster (right) with all CaMKIIs in the active state (**D**) and in the inactive state (**F**). (**G, I**) Top view (left) and side view (right) snapshots of simulations in the 2D system with CaMKII in the active (**G**) and inactive (**I**) states. (**H, J**) Density profiles in the slice of the cluster on membrane along X axis (left) and molecular distribution on membrane along the distance from the center of mass of the cluster (right) with all CaMKIIs in the active state (**H**) and in the inactive state (**J**). Shaded areas indicate standard errors for three independent simulations. The colors of the molecules, trajectories, bar graphs, and distributions are corresponding.

The online version of this article includes the following figure supplement(s) for figure 2:

**Figure supplement 1.** Switching dissociation constants of AMPAR-PSD-95 and NMDAR-CaMKII has little effect on the multiphase morphology both in 3D and 2D systems.

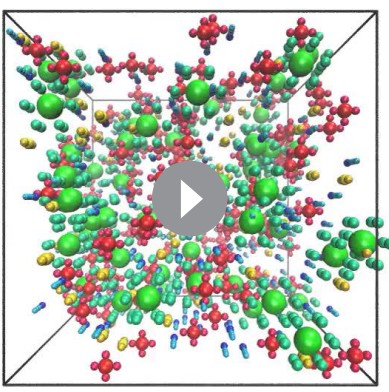

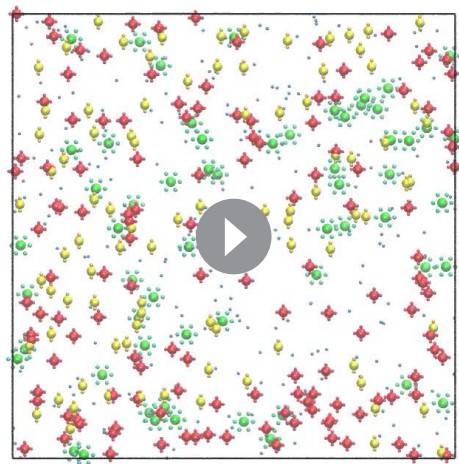

**Video 1.** Phase separation dynamics for a mixture of 135 AMPAR(TARP)$_4$, 240 PSD-95, 90 NMDAR, and 60 active CaMKII in a 3D system.
https://elifesciences.org/articles/106602/figures#video1

**Video 3.** Molecular assembly dynamics for a mixture of 135 AMPAR(TARP)$_4$ and 90 NMDAR embedded on the membrane, 240 palmitoylated PSD-95, and 60 active CaMKII under the membrane in a 2D system, view from the top.
https://elifesciences.org/articles/106602/figures#video3

those interfaces are in contact with each other (*Figure 2G*). From the molecular distribution of the cross sections, it appears that NMDAR clusters more on the inner side and AMPAR clusters more on the outer side (*Figure 2H*).

When we replaced the active CaMKIIs with inactive CaMKIIs and performed a simulation under the same condition, a mixture of AMPARs, NMDARs, and PSD-95s took shape into a slightly larger cluster in about 60ms (*Video 4*, *Figure 2I*). The distribution of these three types of molecules from the cluster center illustrates that NMDARs reside slightly outside of the cluster and AMPAR and PSD-95 form a cluster inside. Inactive CaMKIIs, on the other hand, are all located away from the membrane since they are unable to bind to GluN2Bc (*Figure 2J*). From the results, it is found that with the four PSD components in the 2D system, NMDAR, the active CaMKII, and PSD-95 are clustered on the inner side, while AMPAR(TARP)$_4$ and PSD-95 are clustered on the outer side in the 2D system. The results are consistent with the nanodomain-like structure of the receptor observed in real synapses in several previous studies (*Hosokawa et al., 2021*; *Hruska et al., 2022*; *Li et al., 2021*).

The simulations reproduce experimental features and clearly substantiate that the CaMKII activation induces the multiphasic structure both in the 3D and 2D systems. With inactive CaMKII, which does not attract other molecules, only PSD-95 bridges AMPAR and NMDAR via

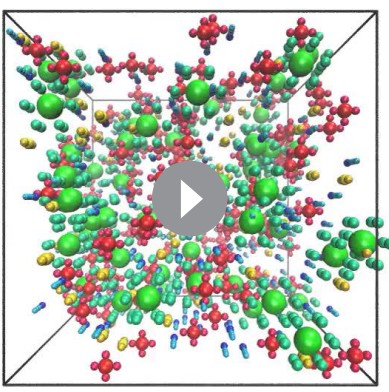

**Video 2.** Phase separation dynamics for a mixture of 135 AMPAR(TARP)$_4$, 240 PSD-95, 90 NMDAR, and 60 inactive CaMKII in a 3D system.
https://elifesciences.org/articles/106602/figures#video2

multivalent specific and non-specific interactions, thus forming a mixed phase. On the other hand, the active CaMKII binds to NMDAR alone. The large volume of CaMKII effectively repels AMPAR from the NMDAR-containing phase to induce the multiphasic architecture in the 3D system.

## Specific and non-specific interactions in 3D/2D systems

In the four-component PSD systems, the 3D and 2D systems exhibit the CaMKII-dependent multi-phase separation in an apparently opposite arrangement of AMPAR-containing phase and NMDAR-containing phase, which is consistent with experiments. The immediate question is why does morphology differ between the 3D and 2D systems. What properties of CaMKII are responsible for such results? Generally, multiphase morphology is ascribed to be shaped by molecular interactions in the condensates, which we explored based on our simulations.

We first investigate multivalent specific interactions between the four protein complexes. To our surprise, the overall type and the number of specific interactions are rather similar between the 3D and 2D systems (*Figure 3A* top, *Figure 3—figure supplement 1A*). Especially, NMDARs keep their interactions with CaMKIIs and PSD-95 nearly the same. This is surprising since the 3D system resulted in a large droplet, whereas the 2D system resulted in relatively small unstable clusters. We note some minor differences. First, the number of PSD-95s that bridge AMPARs and NMDARs is decreased in the 2D system. Second, the number of AMPAR-PSD-95-specific interactions is reduced by about 20% in the 2D system compared to the 3D system.

We then analyze non-specific interactions between four protein complexes in the simulations. In sharp contrast to the specific interactions, we find markedly reduced non-specific interactions in the 2D system compared to the 3D system (*Figure 3A* bottom, *Figure 3—figure supplement 1B*). For example, the number of non-specific contacts between AMPARs and PSD-95, and that between NMDARs and PSD-95 in 2D systems is up to 10 times less than those in the 3D system. The non-specific interaction between NMDARs and CaMKIIs is further reduced. Inherently, the 3D system establishes many more non-specific contacts, whereas the 2D system has more extended configurations with much less contacts.

These two together suggest that the multivalent specific interactions in the 2D system are comparable, while the non-specific interactions in the 2D system are much weaker, compared to the 3D system.

## Roles of CaMKII in 3D/2D systems

The dodecameric CaMKII has a rather high valency for the specific interaction with NMDARs. We investigate how many NMDARs are bound per one CaMKII complex in the simulations (termed 'effective valency', *Figure 3B*). Even though the 2D system organizes smaller clusters compared to the 3D system, we find that each CaMKII in the cluster in the 2D system can bind more NMDARs than in the 3D system. Thus, the effective valency of CaMKII is higher in the 2D system. We note that this increase in the effective valency can be cause or effect of the different morphology in the 3D and 2D systems.

To understand how the high effective valency of CaMKII can be realized in the 2D system, we delve into the localization of four molecular complexes along the normal direction to the membrane both for the cases of the active and inactive CaMKIIs (*Figure 3C*). Both AMPA and NMDA receptors are embedded in the membrane (-2.5 nm $\lesssim z \lesssim$ 2.5 nm). The PDZ2 and GK domains of PSD-95 are located at around (-2 nm $\lesssim z \lesssim$ -5 nm) and (-2 nm $\lesssim z \lesssim$ -8 nm) below

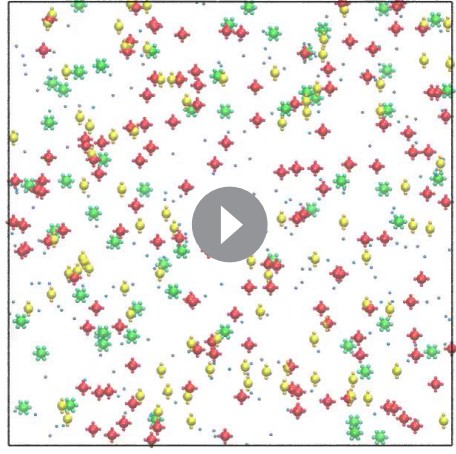

**Video 4.** Molecular assembly dynamics for a mixture of 135 AMPAR(TARP)$_4$ and 90 NMDAR embedded on the membrane, 240 palmitoylated PSD-95, and 60 inactive CaMKII under the membrane in a 2D system, view from the top.

https://elifesciences.org/articles/106602/figures#video4

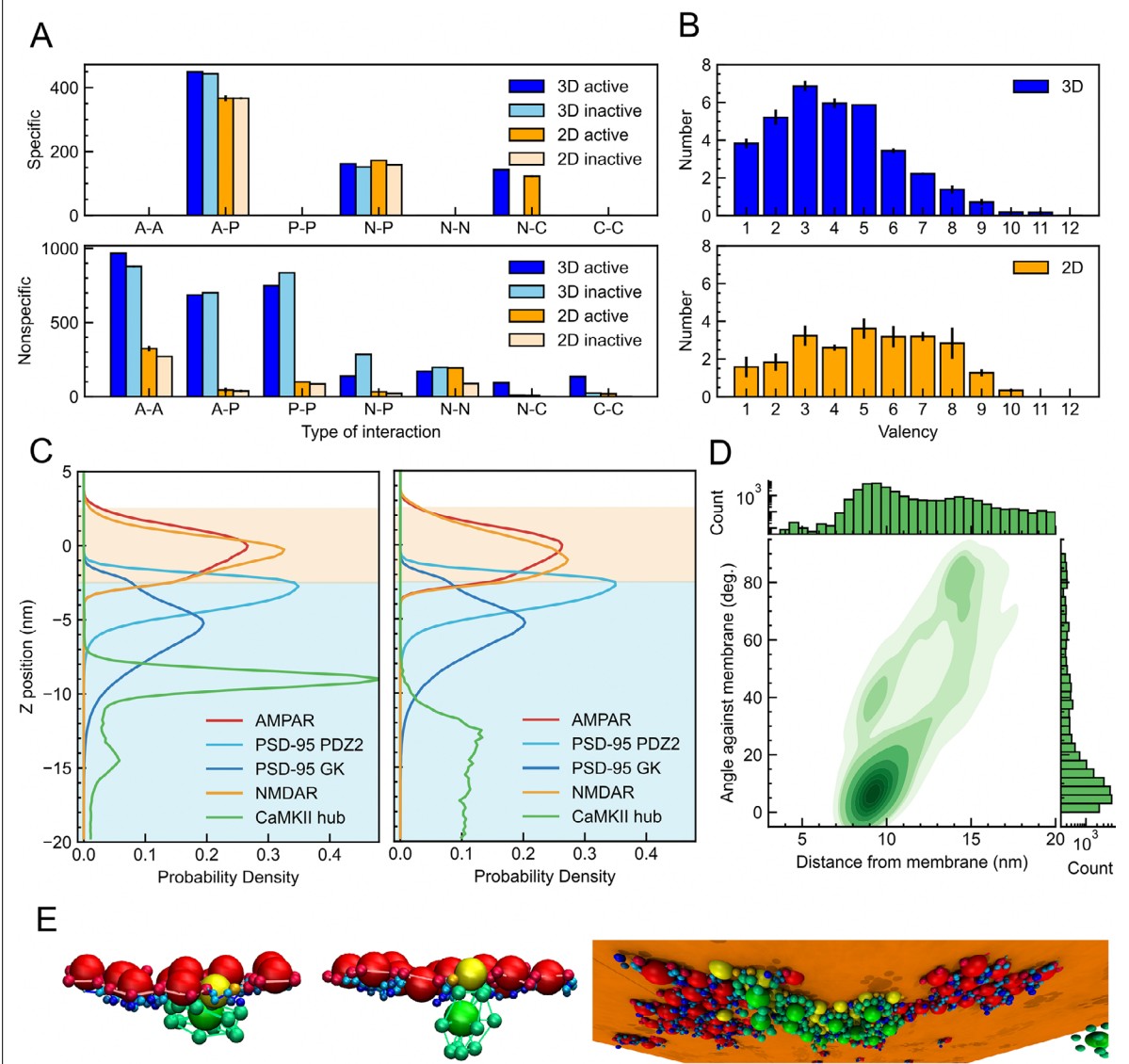

**Figure 3.** Molecular interaction and CaMKII architecture in the four-component PSD assembly. (**A**) Number of specific interaction (top) and non-specific interaction (bottom) with active CaMKII in 3D (blue), inactive CaMKII in 3D (sky-blue) active CaMKII in 2D (orange) and inactive CaMKII in 2D (pale orange). Each index indicates the interaction type where A, P, N, and C represent AMPAR, PSD-95, NMDAR, and CaMKII, respectively. (**B**) Distribution of the number of NMDARs bound to CaMKII molecules in 3D system (top) and 2D system (bottom). (**C**) Protein distribution normal to the membrane plane (the z-coordinate) in the 2D system with the active (left) and inactive CaMKIIs (right). The orange and light blue background represents the membrane and cytoplasmic regions, respectively. (**D**) Two-dimensional kernel density estimation plot of the position and orientation of CaMKII molecules. The horizontal axis represents the distance from the membrane, while the vertical axis shows the angle between the membrane plane and each two hexagonal planes of CaMKII kinases. The height of each histogram and the color of the KDE plot are shown in logarithmic scale. (**E**) Representative snapshots of the 2D systems with active CaMKII. The hexagonal plane of CaMKII is parallel to the membrane plane (left) or perpendicular to the membrane plane (middle). A snapshot of a whole cluster looking from the cytoplasm (right). Orange plane corresponds to the membrane center (z=0).

The online version of this article includes the following figure supplement(s) for figure 3:

**Figure supplement 1.** Number of specific and nonspecific interactions in each system and each CaMKII state.

the membrane, respectively. The active CaMKIIs interacting to NMDARs are distributed around ($-8\,\text{nm} \lesssim z \lesssim -10\,\text{nm}$) (*Figure 3C* left). On the other hand, the inactive CaMKIIs are uniformly distributed 12 nm away from the membrane (*Figure 3C* right). These suggest that, in the 2D system, four molecular complexes are distributed to have a layered organization. More importantly, the active CaMKII occupies ~9 nm below the membrane, which is separated from, but not too far from, the

distribution of the two receptors or PSD-95s. The layered arrangements may enable more NMDARs to access one CaMKII complex.

In the presence of active CaMKII, the distribution of CaMKII appears to have two peaks at approximately –9 nm and –15 nm. Visual inspection suggests that the former and the latter peaks correspond to the horizontal and the perpendicular arrangement of CaMKII. Quantitatively, the angle between a hexagonal surface composed by the six kinase domains of CaMKII and the membrane (*Figure 3D*) shows a maximum frequency at 0°. Therefore, the majority of CaMKII bound to NMDARs is likely to have hexagonal kinase domains attached horizontally to the membrane (*Figure 3E* left). The other peak of CaMKII around –15 nm is widely observed with the angle ranging from 30° to 90°, but not around 0°. This can be regarded as a different binding mode from the previous one, in which the hexagonal plane is perpendicular to the membrane (*Figure 3E* middle). These two different binding modes of CaMKII, horizontal and perpendicular, correspond to the two peaks of CaMKII below the membrane. PSD-95 is a flexible molecule with a string-like structure, whereas CaMKII 12-mer is a molecule with a nearly spherical shape and, moreover, an extremely bulky structure. Although both can interact directly with NMDARs, these differences in flexibility, size, and binding mode are thought to create a hierarchical structure under the membrane and thus provide a firm scaffold for NMDARs (*Figure 3E* right).

Together, we hypothesize that CaMKII has two competing effects on the morphology of multiphase separation; the multi-valency in the specific interactions tends to make the NMDAR-containing phase inside, whereas the large volume of CaMKII that contributes to non-specific interactions tends to put the NMDAR-containing phase outside. The relative balance between the two competing effects is different between the 3D and 2D systems; the non-specific volume effect dominates in the 3D system, whereas the specific multivalent interaction overwhelms the excluded volume interaction in the 2D system.

## Reduction of CaMKII volume can invert multiphasic structure in 3D system

We have shown that (i) the CaMKII activation induces the multiphase separation into AMPAR-containing and NMDAR-containing phases both in the 3D and 2D systems, (ii) the morphology of multiphases in the 3D and 2D systems is apparently opposite, and (iii) relative strengths of specific and nonspecific interactions are sharply different between the 3D and 2D systems. Importantly, CaMKII has a strong impact both on the specific interactions via its multi-valency and on the non-specific interactions via its extraordinarily large volume. These together led us to hypothesize that multi-valency and large volume of CaMKII have competing effects to modulate morphology of multiphases in the 3D and 2D systems. To test our hypothesis, we design a set of artificial systems that modulate multivalent specific interactions or large volume nonspecific interactions of CaMKII.

In the 3D system, nonspecific interactions overwhelm the specific ones to modulate the multiphase morphology. Non-specific interactions of CaMKII are dominated by its large excluded volume. Thus, we design artificial systems with reduced volume of CaMKII keeping other properties unchanged. We carry out simulations by replacing the molecular radii of CaMKII to 2/3 and 1/2 of their original size (*Figure 4A*). With this modification, the volume of CaMKII is reduced by 8/27 and 1/8, respectively. The simulation containing CaMKII with *r*=2/3 of the original radius demonstrates that the AMPAR-containing phase (denoted as A phase) and the NMDAR-containing phase (denoted as N phase) are both separated from the dilute phase to form their respective hemispheres with a shared boundary, which resembles structures called Janus droplets (*Figure 4BC* center). When the radius is halved, CaMKII (*r*=1/2), the N phase is positioned inside of the cluster and nearly completely engulfed by the A phase (*Figure 4BC* right). Namely, reducing the non-specific excluded volume interactions of CaMKII results in the inversions of multiphase morphology in the 3D system. The violin plot in *Figure 4D* illustrates how each of the two types of receptors is distributed starting from the center of mass of the cluster. As the radius is decreased, the peak of AMPARs moved outwards and that of NMDARs moved inwards.

Next, in the 3D system, we reduced specific interactions of CaMKII to see how the phase morphology changes. Specifically, we designed another set of systems with reduced valency of CaMKII. With reduced specific interaction of CaMKII, we originally expected not to observe the opposite morphology, but actually found this did not change the phase morphology (*Figure 4—figure supplement 1*).

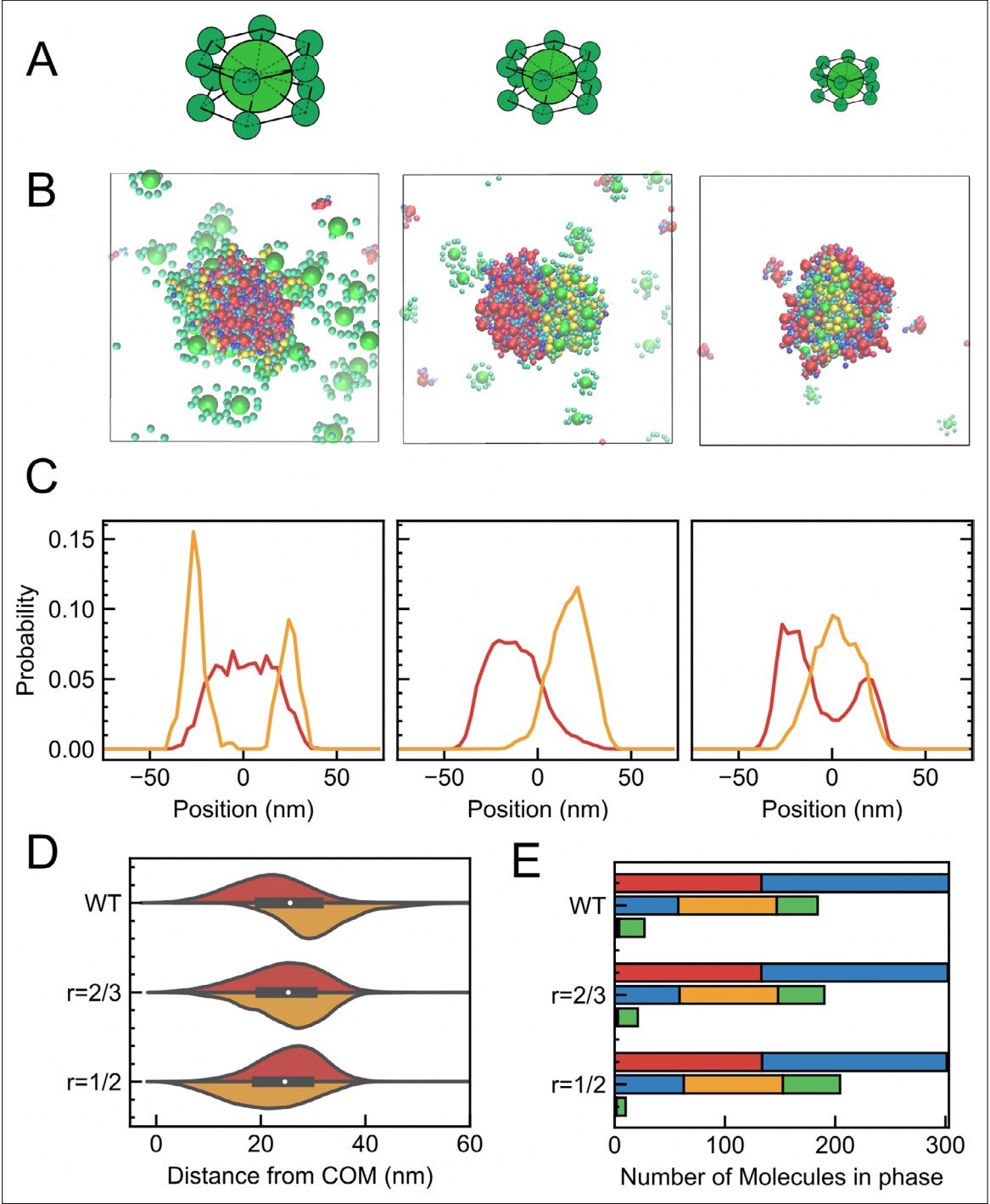

**Figure 4.** Multiphase morphology for CaMKII with reduced volume in the 3D system. (**A**) Schematics of CaMKIIs used in each simulation. Original sized CaMKII (WT) (left), CaMKII (*r*=2/3) with 2/3 of its radius (middle), and CaMKII (*r*=1/2) with 1/2 of its radius (right). (**B**) Snapshots of cross-sectional view with original CaMKII (WT) (left), CaMKII (*r*=2/3) (middle), and CaMKII (*r*=1/2) (right). (**C**) Density profile of AMPAR (red) and NMDAR (orange) in the slice of the cluster along Z axis for CaMKII (WT) (left), CaMKII (*r*=2/3) (middle), and CaMKII (*r*=1/2) (right). (**D**) Molecular distribution of AMPAR (red) and NMDAR (orange) from the center of mass of the cluster (right) of the simulation with CaMKII (WT) (top), that with 2/3 in its radius (middle), and with 1/2 in its radius (bottom). As the molecular radius decreases, the peak of AMPARs shifted outside while that of NMDARs moved close to the cluster center. (**E**) Molecular composition of each phase in the simulation with CaMKII (WT) (top), CaMKII (*r*=2/3) (middle), and CaMKII (*r*=1/2) (bottom). In each case, the three bars indicate the numbers of molecules in the AMPAR-containing phase (top bar), the NMDAR-containing phase (middle bar), and the dilute phase (bottom bar).

The online version of this article includes the following figure supplement(s) for figure 4:

**Figure supplement 1.** Multiphase morphology for CaMKII with reduced valency in the 3D system.

## Reduction of CaMKII valency can invert multiphasic structure in 2D system

Continuing the examination of our hypothesis, we next modulated the setup of 2D simulations to weaken the contribution of specific interactions of CaMKII. We reduced the valency of active kinase domains that can bind to NMDAR; the valency is reduced from the original 12 to 6 (CaMKII (V=6)) and 3e (CaMKII (V=3)) (*Figure 5A*). In the CaMKII with reduced valency, only six or three kinase domains are active, while others are kept as inactive. In the case of CaMKII (V=6), in 60ms, clusters are separated into AMPAR-containing and NMDAR-containing domains (*Figure 5B*). The mean cluster size is smaller than that of the original setup. Some clusters in this condition have a morphology with NMDARs inside and AMPARs outside (dotted arrows, clusters in the middle and bottom right), similar to those obtained in the original CaMKII simulations, whereas others, in contrast, have opposite morphology with NMDARs outside and AMPARs inside (solid arrows, clusters at the bottom). For the case of CaMKII (V=3), there is no clear domain segregation of AMPARs and NMDARs as in the case of 12 or 6 valences, and homogeneous clusters are shaped, almost analogous to the condition with inactive CaMKII (*Figure 5B* right). To gain a more precise understanding of the segregation of receptor domains and their relative positions within clusters in more detail, we examined the distribution of the two types of receptors in clusters above a certain size for each condition, in cross sections at the X=0 line. The receptor distribution of each cluster is normalized by the radius of gyration of the cluster and plotted on the horizontal axis as a relative distance to obtain an accurate estimate of the distribution between clusters of different sizes. In the section with valency V=12, a distinct AMPAR-outward and NMDAR-inward structure is observed. In the case of CaMKII with V=6, the inside-outside separation almost disappears and the NMDARs are evenly distributed both inside and outside the cluster; for CaMKII with V=3, AMPARs are distributed closer to the center and NMDARs are slightly distributed outside (*Figure 5C*). The violin plot also shows that as the valence decreases, AMPARs are distributed closer to the center of the cluster and NMDARs are distributed further away from the center (*Figure 5D*).

On the other hand, when we reduced the volume of CaMKII in the 2D system, the NMDAR phase becomes more rigid as the volume is reduced from the original CaMKII. In addition, the AMPAR and NMDAR clusters do not share a boundary and are completely separated into their own clusters (*Figure 5—figure supplement 1*).

We further attempted to mimic intermediate conditions between 3D and 2D systems in two different manners. First, we applied a weaker membrane constraint in a 2D system. Even when the strength of membrane constraints is reduced by a factor of 1000, NMDARs are located on the inner side when the CaMKII was active, as well as the result in 2D system (*Figure 5—figure supplement 2ABC*). Second, to weaken further the effect of membrane constraints, we artificially altered the membrane thickness from 5 nm to 50 nm, in addition to reducing the membrane constraints by 1000. As a result, NMDAR clusters move to the bottom and surround AMPAR (*Figure 5—figure supplement 2DEF*). In this artificial intermediate condition, both states in which the NMDARs are outside (corresponding to 3D) and in which the NMDARs are inside (corresponding to 2D) are observed, depending on the strength of the membrane constraint.

## Interfacial tensions between each phase substantiate the tendency of multiphase morphologies

In general, the multiphase morphology is determined by the balance of interfacial tensions. When two components, A and B, can form droplets in solution and are mutually immiscible, their mixture can have several morphologies (*Guzowski et al., 2012*). (1) The segregated droplets where two components, A and B, form segregated droplets, which is stable when $\gamma_{AB} > \gamma_A + \gamma_B$ , where $\gamma_A$ is the interfacial tension between A and solution, $\gamma_B$ is that between B and solution, and $\gamma_{AB}$ is that between A and B. (2) The core-shell droplet where an A-containing core droplet is completely engulfed by a shell containing B, when $\gamma_A > \gamma_B + \gamma_{AB}$. (3) The core-shell droplet where a B-containing core droplet is completely engulfed by a shell containing A, when $\gamma_B > \gamma_A + \gamma_{AB}$ . (4) The Janus droplet (partial engulfing) where two droplets are coalesced to have partial A/B interface, for the rest of the cases.

To deepen our understanding of the multiphase morphologies in the current four-component of PSD proteins, we calculated the three interfacial tensions among the three phases (AMPAR-containing phase, NMDAR-containing phase, and dilute phase) in the soluble 3D system. Using the canonical

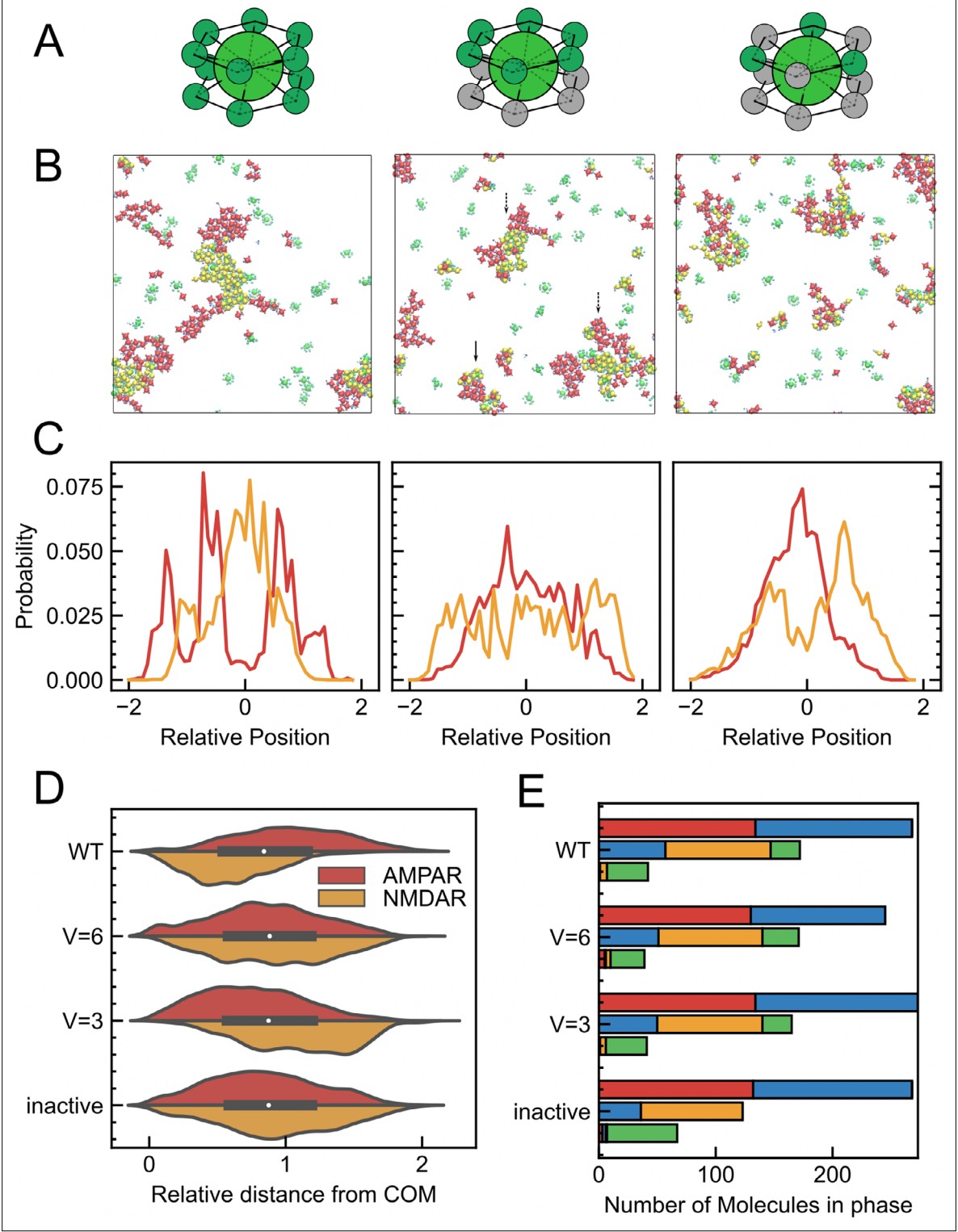

**Figure 5.** Multiphase morphology for CaMKII with reduced valency in the 2D system. (**A**) Schematics of CaMKII used in each simulation. Original 12 valence CaMKII (WT) (left), 6 valence CaMKII (V=6) (middle), and 3 valence CaMKII (V=3) (right). (**B**) Snapshots of cross-sectional view with CaMKII (WT) (left), that with CaMKII (V=6) (middle), and with CaMKII (V=3) (right). (**C**) Density profile of AMPAR (red) and NMDAR (orange) in the slice of the cluster along X axis of the simulation with CaMKII (WT) (left), CaMKII (V=6) (middle), and CaMKII (V=3) (right). As the valency decreases, the architecture of the core-shell cluster shifts between AMPARs and NMDARs. (**D**) Molecular distribution of AMPAR (red) and NMDAR (orange) from the center of mass of the cluster (right) of the simulation with CaMKII (WT) (top), that with CaMKII (V=6) (top middle), that with CaMKII (V=3) (bottom middle), and that

*Figure 5 continued on next page*

*Figure 5 continued*

with inactive CaMKII (bottom). As the molecular valence decreases, the peak of AMPARs shifts inside while that of NMDARs moves far from the cluster center. (**E**) Molecular composition of each cluster in the simulation with CaMKII (WT) (top), that with CaMKII (V=6) (top middle), that with CaMKII (V=3) (bottom middle), and that with inactive CaMKII (bottom). The three bars indicate the number of molecules in each of the phases containing AMPARs (top bar), NMDARs (middle bar), and others (bottom bar), respectively. The composition of phase with NMDAR does not change drastically between different CaMKII sizes.

The online version of this article includes the following figure supplement(s) for figure 5:

**Figure supplement 1.** Multiphase morphology for CaMKII with reduced volume in the 2D system.

**Figure supplement 2.** Alleviating membrane constraints controls the partitioning state of receptors.

ensemble, we prepared a three-dimensional slab system (*Figure 6A*) and estimated the interfacial tension of each interface based on the Kirkwood-Buff equation (*Kirkwood and Buff, 1949*) with varying size of CaMKIIs. In the system containing CaMKII (WT), the interfacial tension between the AMPAR-containing phase and the dilute phase ($\gamma_A$) is larger than the sum of the other two, the interfacial tension $\gamma_N$ between the NMDAR-containing phase and the dilute phase and $\gamma_{AN}$ between the A and N phases, $\gamma_A > \gamma_N + \gamma_{AN}$ (*Figure 6BC*). Thus, the core-shell morphology with the A phase at the

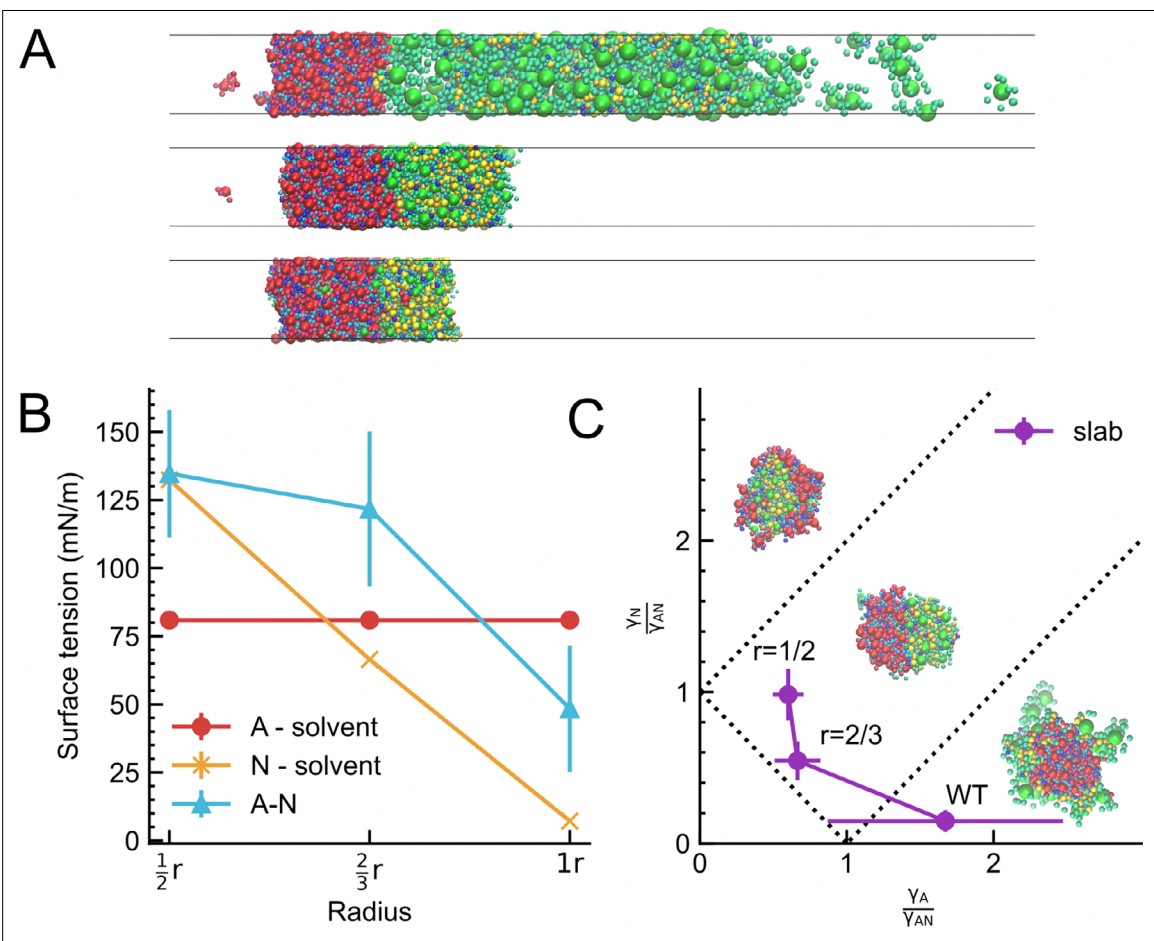

**Figure 6.** Interfacial tensions explain the multiphase morphology in 3D system. (**A**) Representative snapshots of 3D slab simulation used in the measurement of interfacial tension between A (AMPAR-containing, red) and N (NMDAR-containing, green) phases, with CaMKII (WT) (top), CaMKII (*r*=2/3) (middle), and CaMKII (*r*=1/2) (bottom). (**B**) Interfacial tensions between the A and dilute phases ($\gamma_A$, red line), between the N and dilute phases ($\gamma_N$, orange line) and between the A and N phases ($\gamma_{AN}$, cyan line) with error bars at each condition. (**C**) Phase diagram on the $\gamma_A/\gamma_{AN}$ and $\gamma_N/\gamma_{AN}$ plane. Dashed lines represent theoretical borders of multiphase morphology.

The online version of this article includes the following figure supplement(s) for figure 6:

**Figure supplement 1.** Fixed molecular composition does not change the relative relationship between the interfacial tensions between the phases in 3D system.

core and N phase in the shell should be most stable. Indeed, this is what we observe in the simulation. As the volume of CaMKII decreases, we find both $\gamma_N$ and $\gamma_{AN}$ increase, whereas $\gamma_A$ is unaffected (*Figure 6BC*). With the CaMKII ($r=2/3$), our estimate satisfies $\gamma_A < \gamma_N + \gamma_{AN}$ and $\gamma_{AN} < \gamma_A + \gamma_N$ as the Janus droplet appears (*Figure 6BC*). With further reduced volume of CaMKII, $\gamma_N$ increases to approach the border to the reversed core-shell morphology with the N phase at the core and the A phase in the shell. However, while we observe nearly core-shell arrangements with the N phase at the core with CaMKII ($r=1/2$), the estimated tensions do not satisfy $\gamma_N > \gamma_A + \gamma_{AN}$. This apparent discrepancy may be partly due to the relatively small size of the simulated system, whereas the theory rests on a macroscopic argument. In addition, the numerical estimate of the interfacial tension, especially $\gamma_{AN}$, is inherently difficult and may contain a non-negligible error. Yet, the overall trend of the interfacial tensions explains qualitatively the observed changes in the multiphase morphologies, supporting our interpretation of the underlying physical mechanism.

## Discussion

In this study, we investigate the multiphase morphology of biomolecular condensates induced by multivalent interactions and excluded volume interactions in 3D solution and 2D membrane-bound systems, using PSD proteins as a model system. Utilizing a mesoscale model which can simulate molecules on a comparatively large scale in both time and space, we first show that the CaMKII activation induces the multiphase separation and that four components of PSDs form reversed multiphase morphologies between the 3D solution system and the 2D system on and under the membrane. Analyzing the molecular interactions, we identify that, while the 3D system contains a vast number of non-specific volume interactions, the 2D system is dominated by specific multivalent interactions. Especially in the PSD protein system, the excluded volume interaction of CaMKII drives the NMDAR/CaMKII-containing phase in the shell of the core-shell morphology in the 3D system, whereas multivalent specific interaction of the active CaMKII stabilizes the NMDAR/CaMKII-containing domain at the core of the 2D system. These results suggest that the layered structure of the PSD created by the membrane geometry reduces the effective volume interaction and thus increases the relative contribution of multivalent interactions in 2D systems.

The segregation of nanodomains containing NMDAR or AMPAR can hold significant functional significance in synaptic functions. The clustering of receptors is generally considered essential for efficient neurotransmitter reception from presynaptic terminals (*Fukata et al., 2024*; *Nair et al., 2013*). Nanodomains of NMDARs and AMPARs are separately aligned with synaptic vesicle release sites in presynaptic cells via different assembly proteins in synaptic clefts and are used in distinct contexts. The current study shows that the NMDAR-containing nanodomains tend to be located near the center of the postsynaptic membrane, whereas the AMPAR-containing nanodomains are at the periphery, which is consistent with recent in vivo observations of receptor locations (*Hosokawa et al., 2021*; *Hruska et al., 2022*; *Li et al., 2021*). While these in vivo results contain additional scaffold and cytoskeletal elements omitted in our model, such as SAPAP, Shank, and Homer, nearly all proteins in the middle and lower layers of the PSD associate directly or indirectly with PSD-95 in the upper PSD layer. Consequently, it is probable that other scaffold proteins contribute to the mobility of AMPAR-containing and NMDAR-containing nanodomains indistinguishably. They may increase the stability of the AMPAR and NMDAR clusters but are unlikely to have a distinct effect to reverse the phase-separation phenomenon. Notably, our simulations contain only proteins in the PSD, but neither presynaptic nor synaptic cleft proteins. Therefore, the results imply that the postsynaptic proteins alone may be sufficient to form the nanodomain morphology observed in vivo.

While the estimated interfacial tension well explains the multiphase morphology in the 3D system, the same approach cannot be used in 2D because the 2D system does not show stably separated phases, but only nanoscale dynamic clusters (*Yamada and Takada, 2023*). For such clusters, the interfacial tension is ill-defined. However, we can still rationalize the cluster morphology on the membrane by focusing on its interfacial properties. There are reasons to speculate that the domain boundary cost between domains containing NMDARs/CaMKII and the dilute phase in 2D systems is greater than that containing AMPARs. First, in the light of the fluctuation in the copy numbers of receptors in the largest cluster, the NMDAR clusters show markedly smaller fluctuations and exhibit step-wise growth similar to that in the 3D system (*Figure 2AB*). According to the capillary-wave theory, the larger the interfacial fluctuation is, the smaller the interfacial tension is (*Buff et al., 1965*; *Ismail et al., 2006*;

*Lacasse et al., 1998*; *Semenov, 1994*; *Sides et al., 1999*). Thus, the NMDAR-containing domain can be regarded to have a larger domain boundary cost than that of the AMPAR domain. Furthermore, the number of multivalent interactions of NMDAR with either PSD-95 or CaMKII was almost the same between 3D and 2D systems, whereas that between AMPAR and PSD-95 was significantly reduced (*Figure 3A*). This is in harmony with the view that the decline of domain boundary cost of AMPARs with the dilute phase in 2D system relative to that of NMDARs. At first glance, this seems to be due to differences in the dissociation constants of the specific interactions. However, modulating the $K_d$ of the AMPAR-PSD-95 and NMDAR-CaMKII-specific interactions did not change the multiphase morphology, and thus domain boundary cost is not strongly dependent on $K_d$ of the molecules (*Figure 2—figure supplement 1*). Instead, the multivalence of the interaction is the primary determinant of the interfacial cost and thus the domain morphology below the membrane. NMDARs migrate outwards when the multivalent interaction is disturbed by a decrease in the CaMKII valence.

Generally, the phase separation can be explained by the Flory-Huggins theory and its extensions: phase separation can be favored by the difference in the effective pairwise interactions in the same phase compared to those across different phases and is disfavored by mixing entropy. The effective interactions contain various molecular interactions, including direct van der Waals and electrostatic interactions, hydrophobic interactions, and purely entropic macromolecular excluded volume interactions. For the latter, Asakura-Oosawa depletion force can drive the phase separation. Furthermore, the demixing effect was explicitly demonstrated in previous simulations and field theory (*Pal et al., 2021*). Importantly, we note that the effective pairwise interactions scale with the coordination number $z$. The coordination number is a clear and major difference between 3D and 2D systems. In 3D systems, large $z$ allows both relatively strong few specific interactions and many weak non-specific interactions. While a single specific interaction is, by definition, stronger than a single non-specific interaction, the contribution of the latter can have a strong impact due to its large number. On the other hand, a smaller $z$ in the membrane-bound 2D system limits the number of interactions. In the case of limited competitive binding, specific interactions tend to be prioritized compared to non-specific ones. In fact, *Figure 3A* clearly shows that the number of specific interactions in 2D is similar to that in 3D, while that of non-specific interactions is dramatically reduced in 2D. In the current PSD system, CaMKII is characterized by large valency and large volume. In the 3D solution system, non-specific excluded volume interactions drive CaMKII to the outer phase, while this effect is largely reduced in 2D, resulting in the reversed multiphase.

Modulation of the balance between specific and non-specific interactions via the membrane geometry should not be limited to the PSD proteins, but can be a general phenomenon. Domain-wise specific interactions tend to be relatively strong (per domain) and have strict, often one-to-one, stoichiometry. Non-specific, or more broadly, less specific, interactions tend to be weak and not to have clear stoichiometry. In the 3D solution systems, each protein can interact with many other proteins via specific or non-specific interactions. On the other hand, membrane-associated protein systems can have less interaction pairs per protein inherently due to their reduced dimensionality. Limited pairs should be dominated by strong specific interactions, thus reducing the contribution of non-specific interactions. Thus, compared to 3D solution systems, multivalency in specific interactions should be more important on and beneath the membrane. In fact, recent observations have revealed that multiphase separation by LLPS is more likely to occur at the two-dimensional solid-liquid interface than in a three-dimensional liquid droplet (*Gong et al., 2024*). In a pseudo-2D system, specific interactions may be effective and facilitate multiphase separation.

The current study elucidates the primary role of specific multivalent interactions on and beneath the membrane, which was previously illustrated for the Nephrin/Nck/N-WASP complex by Rosen's group and others (*Case et al., 2019b*; *Li et al., 2012*). The Nephrin/Nck/N-WASP complex exhibits multivalent interactions through SH3 domains and PRMs. This complex not only induces phase-separated macroscopic droplets in 3D in vitro solution systems (*Li et al., 2012*) but also shows 2D in vitro phase-separated clusters on membranes driven by increasing valency (*Case et al., 2019b*). These two-dimensional clusters exhibit fluid properties, as in the 3D droplets. Our current research further expands on these concepts, elucidating the primacy of specific multivalent interactions in the 2D system, compared to the relative importance of non-specific volume interactions in the 3D system. Especially, this difference in the balance between specific and non-specific interactions causes a reversal of the multiphase morphology in the case of the four-component PSD system.

Physical mechanisms of demixing into multiphase droplets, especially the role of multivalent interactions therein, have been intensively studied by Brangwynne's group (*Rana et al., 2024*). In vitro 3D solution systems that can dynamically change the multivalency via oligomerization are designed to demonstrate that the increase of valency by protein oligomerization can enhance multiphase immiscibility leading to core-shell morphology (*Rana et al., 2024*). Notably, some of the tested systems use extraordinarily high valence (24- or 60-folds) to realize the demixing. The current study reveals that specific multivalent interactions, relative to non-specific volume interactions, operate more efficiently in 2D systems. Therefore, we hypothesize that a similar oligomerization-based system may form immiscible multiphase clusters at lower valences on and beneath the membrane.

While our study provides a quantitative framework for understanding multiphase liquid phase separation at the PSD, several limitations should be noted. First, our results did not provide direct insights to physiological conditions, such as ion concentrations. Since such factors are implicitly implemented in our model, it is difficult to estimate these effects individually. This suggests the need for future implementation of environmental factors and validation under a broader range of in-vivo-like settings. In this context, we also do not explicitly account for downstream phosphorylation events. Although such proteins are not included in the current components, they will regulate PSD-95, affecting its binding valency or diffusion coefficient. This is a subject worthy of future research. Second, parameter calibration contains some uncertainty. Previous in vitro study results used for parameter validation are at relatively high concentrations for phase separation, which may shift critical thresholds compared to that in in vivo environments. Also, since the number of molecules included in the model is small, the difference of a single molecule could result in a large error during this validation process. Third, we estimated all the diffusion coefficients from the Einstein-Stokes equation, which may oversimplify membrane-associated dynamics. Applying the Saffmann-Delbrück model to membrane-embedded particles would be desired, although the resulting diffusion coefficients remain broadly of the same order of magnitude. These limitations highlight the need for further research, yet they do not undermine the core significance of the present findings in advancing our understanding of multiphase morphologies.

Biomolecular condensates are ubiquitous in cells, most notably in the nucleus and under the membrane. Membrane-associated biomolecular condensates are linked to signal transduction, membrane shape manipulation, stepwise transport of molecules and vesicle-membrane tethering by wetting (*Day et al., 2021*; *Jaqaman and Ditlev, 2021*; *Kaksonen and Roux, 2018*; *Kim et al., 2024*). Most cases, including Nephrin/Nck/N-WASP, Galectin/Saccharide, and Grb2/Sos1 systems, are mediated by multidomain proteins, which thus possess specific multivalent interactions for LLPS (*Case et al., 2019a*). Condensate formations in the nucleus are often linked to activation/inactivation of local chromatin regions associated with epigenetic marks (*Gibson et al., 2019*; *Wang et al., 2019*). In contrast to the membrane-associated cases, many nuclear proteins/RNAs that form condensates in 3D solutions are characterized by full of intrinsically disordered regions (IDRs)/heterogeneous folds including transcription factors, RNA polymerases, and ribosomal RNAs in nucleolus. Interactions therein are dominated by less specific interactions of disordered polymers. Our current study points out that specific multivalent interactions play major roles in the membrane systems, whereas the non-specific interactions that do not have strict stoichiometry are relatively important in the 3D solution system. We emphasize that this difference in the relative importance is the direct outcome of the different dimensionality, and thus is not limited to PSD systems, but can be general. We speculate that membrane-associated condensates and condensates in nuclei may have evolved differential interaction features under distinct pressures resulting from their differences in dimensionality.

Our mesoscale model approach not only provides a better understanding of the PSD and synaptic plasticity, but also allows prediction of how the arrangement and structure of multi-component biomolecular condensates, driven by multivalent interactions, affects the LLPS in various systems.

## Materials and methods

Methods are briefly summarized here; a detailed version is provided in Appendix. Representative trajectory files, checkpoint folders, and python scripts are accessible in the Zenodo repository 'Data for Multiphase separation in postsynaptic density regulated by membrane geometry via interaction valency and volume' (https://doi.org/10.5281/zenodo.15679128).

## Molecular representation

We represented each protein domain as a spherical particle. The radius of each spherical domain $R_n$ and the diffusion coefficient D are calculated by the formula:

$$R_n = 0.224\, N^{0.392} \tag{1}$$

$$R_h = 1.45\, R_n \tag{2}$$

$$D = \frac{k_B T}{6\pi\eta R_h} \tag{3}$$

The viscosity of the cytoplasmic domain in the 3D system and the 2D system is determined based on the viscosity of water at 300 K (0.89 cP); the viscosity of the protein domains on membrane in the 2D system is set to 10 times the viscosity of the cytoplasmic fluid (8.9 cP) to reduce the calculation time required for assembly.

## Basic simulation method

All simulations were performed using ReaDDy 2 (*Hoffmann et al., 2019*; *Schöneberg and Noé, 2013*). The overdamped Langevin equation is used for the equation of motion. The calculation time-step and temperature of all the simulations is set to 0.25 ns and 300 K, respectively, for all the systems.

Every particle is subject to the following total potential energy function:

$$V_{total} = V_{bond} + V_{angle} + V_{dihedral} + V_{spe} + V_{nonspe} + V_{system} \tag{4}$$

The bonds between AMPAR and TARP, NMDAR and GluN2Bc, and the hub and kinase domains of CaMKII are set as harmonic bonds with a spring constant of 10 kJ/mol/nm$^2$. In contrast, all domain linkers of PSD-95 are represented by modified flexible Gaussian polymer chains, which have a soft spring constant determined by the length of linker amino acids.

For the 2D system, as a system potential, we apply an implicit membrane potential

$$V_{system} = \sum \begin{cases} 0, & |z| < D_{mem2} \\ \frac{1}{2} k_{mem} \left(|z| - D_{mem2}\right)^2, & |z| \geq D_{mem2} \end{cases} \tag{5}$$

for transmembrane domains, where $k_{mem}$ is set to 10 kJ/mol/nm$^2$ and the half membrane thickness $D_{mem2}$ is set to 2.5 nm. For cytoplasmic domains, we apply

$$V_{system} = \sum \begin{cases} 0, & z < -D_{mem2} \\ \frac{1}{2} k_{mem} \left(z + D_{mem2}\right)^2, & z \geq -D_{mem2} \end{cases} \tag{6}$$

Other detailed potentials (angle, dihedral, specific, and nonspecific interaction potentials) are described in Supporting Information.

## Simulation in the 3D system

In the 3D system containing four kinds of protein complexes, as an initial configuration, we placed 135 molecules of DsRed2(TARPc)$_4$, 240 molecules of PSD-95, 90 molecules of eqFP670(GluN2Bc)$_2$, and 60 molecules of CaMKII randomly avoiding their mutual overlaps, in a 155.362 × 155.362 × 155.362 nm cubic box with the periodic boundary condition. The amount of PSD-95 is determined so that the total amount of PDZ domains is equal to the amount of its binding clients (sum of TARPc and GluN2Bc). This setup theoretically allows the PSD-95 and the two types of receptors to bind without excess or deficiency, while the amount of CaMKII is in excess relative to the possible binding sites of NMDARs. To investigate the morphology of the two phases, we adopt a cubic box system, rather than an elongated box; the concentrations of the dilute phases may be subject to artifacts from the finite size and thus not be very accurate as an estimate of critical concentration of LLPS. Then, three independent simulations at 300 K with different stochastic forces are repeated for $8 \times 10^7$ MD timesteps.

## Simulation in the 2D system

In the 2D system containing the same amount of four kinds of protein complexes as in 3D setup, AMPAR(TARP)$_4$ and NMDAR(GluN2Bc)$_2$ are embedded to the membrane, PSD-95 is tethered to the membrane via the palmitoyl group, and CaMKII is in the cytoplasm beneath the membrane. In a 500 × 500 × 500 nm cubic box (the periodic boundary condition), we set the plane at the center of the box (Z=0) as a center of the postsynaptic membrane and apply a potential energy for the membrane. Transmembrane proteins AMPARs and NMDARs and the palmitoyl group of PSD-95 are assumed to diffuse only on the membrane due to this membrane potential. Other domains of PSD-95 and CaMKII can move around in the cytoplasm below the membrane (Z < -2.5 nm). Starting from the randomly placed initial conditions, simulations at 300 K are performed for $8 \times 10^7$ MD time steps.

## Measurement of interfacial tension in the 3D slab system

For measurement of the interfacial tension of each phase, we first conducted the simulation in the 50 × 50 × 500 nm periodic box for $4 \times 10^6$ MD timestep, confining the molecules written in *Appendix 1— table 4* for each condition into a small region of 50 × 50 × 100 nm by applying a confinement potential. The number of molecules in the slab box is chosen to meet the same stoichiometry of each molecule as the simulation in 3D cubic box. Then, we remove the confinement potential and continue the simulation for $2 \times 10^7$ MD timesteps in an elongated 50 × 50 × 1500 nm box with the periodic boundary condition. From the pressure tensor obtained in the simulations, we estimate interfacial tension value between respective condensed and dilute phases, using Kirkwood-Buff equation (*Ismail et al., 2006*; *Kirkwood and Buff, 1949*) as follows:

$$\gamma = \frac{1}{2} L_z \left\{ P_{zz} - \frac{1}{2} \left( P_{xx} + P_{yy} \right) \right\} \tag{7}$$

The amount of CaMKII in each condition is corresponding to the result obtained from simulations in a cubic box. It should be noted that we have confirmed that the value of interfacial tension does not change dramatically even when the CaMKII content was fixed (*Figure 6—figure supplement 1*).

## Acknowledgements

We thank Yutaka Murata, Tsuyoshi Terakawa, and Evangelia Pantatosaki for insightful discussions. This work was supported by JSPS KAKENHI grants 24KJ1449 (to RY), 21H02441 (to ST), and 24K01991 (to ST).

## Additional information

### Funding

| Funder | Grant reference number | Author |
| --- | --- | --- |
| Japan Society for the Promotion of Science | 24KJ1449 | Risa Yamada |
| Japan Society for the Promotion of Science | 21H02441 | Shoji Takada |
| Japan Society for the Promotion of Science | 24K01991 | Shoji Takada |

The funders had no role in study design, data collection and interpretation, or the decision to submit the work for publication.

### Author contributions

Risa Yamada, Conceptualization, Resources, Data curation, Formal analysis, Funding acquisition, Validation, Investigation, Visualization, Methodology, Writing – original draft, Writing – review and editing; Giovanni B Brandani, Supervision, Writing – original draft, Writing – review and editing; Shoji Takada, Supervision, Funding acquisition, Writing – original draft, Project administration, Writing – review and editing

## Author ORCIDs
Risa Yamada https://orcid.org/0009-0000-3573-4045
Giovanni B Brandani http://orcid.org/0000-0003-3379-0187
Shoji Takada https://orcid.org/0000-0001-5385-7217

Reviewer #2 (Public review): https://doi.org/10.7554/eLife.106602.3.sa1
Reviewer #3 (Public review): https://doi.org/10.7554/eLife.106602.3.sa2
Author response https://doi.org/10.7554/eLife.106602.3.sa3

## Additional files

### Supplementary files
MDAR checklist

### Data availability
Representative trajectory files, checkpoint folders, and python scripts are accessible in the Zenodo repository (https://doi.org/10.5281/zenodo.15679129).

The following dataset was generated:

| Author(s) | Year | Dataset title | Dataset URL | Database and Identifier |
|---|---|---|---|---|
| Risa Y | 2025 | Data for Multiphase separation in postsynaptic density regulated by membrane geometry via interaction valency and volume | https://doi.org/10.5281/zenodo.15679129 | Zenodo, 10.5281/zenodo.15679129 |

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

## Appendix 1

## Molecular representation

As in our previous study, we represent each protein domain as a spherical particle. In the 2D system, AMPAR molecules are assumed to be composed of a homotetramer of the GluA2 subunit, which together is modeled as one particle (PDB ID: 3KG2, N = 3292 residues). In the 3D system, we substitute the particle representing the AMPAR with a soluble fluorescent protein called DsRed2, also in tetrameric structure (*Hosokawa et al., 2021*). The TARP γ-2 subunit (N = 211 residues) is modeled as a single spherical particle. Throughout the simulations, it is assumed that the four TARP particles are always linked to one AMPAR (or DsRed2) and never dissociate from it. Although the particles at the center in the 3D system (DsRed2) differ from those in the 2D system (AMPAR), these AMPAR-TARP complexes are referred to as AMPAR(TARP)$_4$ in both systems merely for simplicity. The strength of the potential being applied to each molecule is the same for all of them, while the distance of the interaction depends on the size of the molecule due to the different sizes of the molecules.

As in our previous work, PSD-95 is represented as three particles representing three PDZ domains (PDZ1, PDZ2, and PDZ3, all N = 86 residues), one particle for SH3 domain (N = 70 residues), and one for GK domain (N = 165 residues) connected by Gaussian polymer chains in the order (*Hafner et al., 2015*; *Zeng et al., 2019*). Additionally, in the 2D system, small particles representing the first 5 amino acids with two cysteine residues in PSD-95, where palmitoylated on membrane, are added as the N-terminus (N = 5 residues) and tethered to the postsynaptic membrane to mimic the palmitoylated structure (*Topinka and Bredt, 1998*).

In the 2D system, NMDAR is assumed to consist of a heterotetramer (PDB ID: 4PE5, N = 3290 residues) in which two GluN1s and two GluN2Bs are alternatively arranged (*Paoletti et al., 2013*; *Ulbrich and Isacoff, 2008*). In NMDAR, only GluN2B has a PDZ-binding motif in its C-terminus and can specifically bind to the PDZ domains of PSD-95 (*Cui et al., 2007*). Thus, each of the C-termini of GluN2B (denoted as GluN2Bc) is modeled as a single particle (residues 1226 to 1482, N = 256 residues), whereas the rest, two GluN2Bs except its C-terminus and two GluN1s together are represented as a single particle (the core of NMDAR). Since the GluN2B subunits are alternatively arranged in NMDAR, the entire NMDAR molecule is modeled as a linear structure consisting of three particles; two GluN2Bc's sandwich the core of NMDAR. In the 3D system, we substitute a particle representing the NMDAR with a soluble fluorescent protein called eqFP670, which has a dimeric structure (*Hosokawa et al., 2021*). Although the core of NMDAR in the 3D system (fluorescent protein, eqFP670) differs from the core of NMDAR in the 2D system, NMDARs are referred to as NMDAR(GluNB2c)$_2$ in both systems for brevity.

CaMKII normally exists in a dodecamer. Its monomer consists of a kinase domain (catalytic domain), an autoinhibitory domain, a variable segment, and a self-association domain (*Myers et al., 2017*). In our mesoscopic model, we assume that CaMKII forms a dodecamer, with a hub domain, which is an association of 12 self-association domains, and 12 kinase domains of each monomer connected by a linker. Among the many CaMKII isoforms, we use CaMKIIα, which is thought to be mainly involved in synaptic plasticity and long-term potentiation. The size of each domain is set to 11 nm in diameter for the hub domain and 4.5 nm for the kinase domain based on previous studies (*Myers et al., 2017*; *Tsujioka et al., 2023*). The CaMKII molecule may undergo autophosphorylation in the presence of calmodulin activated by Ca$^{2+}$. Autophosphorylation results in sustained catalytic activity of the kinase domains, and this state is called an active state. At this state, the length of the linker consisting of the variable segment and autoinhibitory domain is extended to 3 nm, due to the binding of calmodulin. In addition, the binding of calmodulin physically separates the kinase domain from the self-association domain, allowing GluN2Bc to bind to the autoinhibition domain of CaMKII (*Bayer et al., 2001*). Contrarily, when calmodulin dissociates from the autoinhibitory domain by dephosphorylation, it leaves the activated state CaMKII becomes inactive. The dissociation of calmodulin shortens the length between the hub domain and the kinase domain, resulting in a compact structure and loss of binding ability to Glun2Bc. To estimate the effect of the active and inactive states of CaMKII on receptor assembly, we model active and inactive CaMKII as distinct structures and conduct an independent simulation in each state. The linker length of active CaMKII is assumed to be 3 nm and that of inactive one to be 0 nm.

The radius of the spherical domain $R_n$ and the diffusion coefficients D are calculated by the formula:

$$R_n = 0.224\, N^{0.392} \tag{A1}$$

$$R_h = 1.45\, R_n \tag{A2}$$

$$D = \frac{k_B T}{6\pi\eta R_h} \tag{A3}$$

For domains of CaMKII, the diffusion coefficient is calculated based on the radius obtained from the 3D Cryo-EM structure, not on the number of amino acid residues. The viscosity of the cytoplasmic domain in the 3D system and the 2D system is determined based on the viscosity of water at 300 K (0.89 cP); the viscosity of the protein domains on membrane in the 2D system is set to 10 times the viscosity of the cytoplasmic fluid (8.9 cP) to reduce the calculation time required for assembly.

## Basic simulation method

All simulations are performed using ReaDDy 2 (*Hoffmann et al., 2019*; *Schöneberg and Noé, 2013*). The overdamped Langevin equation is used for the equation of motion. The calculation timestep and temperature of all the simulations are set to 0.25 ns and 300 K, respectively, for all the systems.

Every particle is subject to the following total potential energy function:

$$V_{total} = V_{bond} + V_{angle} + V_{dihedral} + V_{spe} + V_{nonspe} + V_{system} \tag{A4}$$

The bonds between AMPAR and TARP, NMDAR and GluN2Bc, and the hub and kinase domains of CaMKII are set as harmonic bonds with a spring constant of 10 kJ/mol/nm$^2$. For AMPAR(TARP)$_4$, we use additional harmonic bond between the nearest TARP molecules in the complex to reduce the flexibility in the distance between neighboring TARPs. In contrast, all domain linkers of PSD-95 are represented by modified flexible Gaussian polymer chains ($V_{GPC}$).

$$V_{GPC} = \sum \frac{3k_B T}{2Nl^2}(r - r_0)^2 \tag{A5}$$

Here, $r$ is the distance between the geometric centers of neighboring domains, and $r_0$ is the sum of radii of neighboring domains. Thus, $r - r_0$ represents a distance between spherical surfaces of neighboring domains. It should be noted that the potential is slightly different from standard Gaussian polymer chain model.

For AMPAR(TARP)$_4$, NMDAR, and CaMKII, the angle potential ($V_{angle}$) and dihedral potential ($V_{dihedral}$) are applied in addition to the harmonic bonds to reproduce the accurate shape of each molecular complexes.

$$V_{angle} = \sum k_\theta (\theta - \pi)^2 \tag{A6}$$

$$V_{dihedral} = \sum k_\phi (1 + cos(\phi - \phi_0)) \tag{A7}$$

where $\theta$ and $\phi$ are angles between two consecutive bonds and dihedral angles of three consecutive bonds, respectively. The value of the spring constant $k_\theta$ is set to 10 kJ/mol/nm$^2$ for all the complexes to maintain their proper geometry. $k_\phi$ is set to 10 kJ/mol/nm$^2$ for any dihedral potential used in the AMPAR(TARP)$_4$ complex and CaMKII, and $\phi_0$ is set to 0 rad. We apply the dihedral potential to constrain the four TARP molecules in the same plane, thereby increasing their constraint on the membrane plane. For GluN2Bc-NMDAR-GluN2Bc, only the angular potential $V_{angle}$ is applied. The

structure of CaMKII can be characterized as a shape that two planar hexagons, each consisting of six kinase domains, are arranged on the top and bottom planes. The distance between the top and the bottom monomeric kinase domains of the CaMKIIα is estimated to be about 5.9 nm (*Buonarati et al., 2021*); based on this distance, harmonic bonds, angle, and dihedral potentials were applied so that the molecules forming hexagons on the top and bottom surfaces, respectively, are in the same plane.

As in our previous work, we apply two types of interaction between particles, specific and nonspecific interactions, to represent liquid-liquid phase separation by PSD condensate.

The specific interaction $V_{spe}$ models one-to-one stoichiometric binding between specific protein domains. The specific binding is mimicked by a virtual bond generated by virtual reactions, defined by association and dissociation events. The association reaction occurs with a microscopic rate constant $k_{on}$ when the distance between interacting domains is within a threshold (set as $r_0$, the sum of the radii of interacting particles). The occurrence of association is tried with Poisson probability $P = 1 - \exp(-k_{on}\Delta t)$ in every timestep. After the association event, harmonic interaction is established only for bound interactive domains:

$$V_{\text{spe}} = \sum k_{\text{spe}}(r - r_0)^2 \tag{A8}$$

where $k_{\text{spe}}$ was set to 10 kJ/mol/nm², which is the same as that for other harmonic bonds. Dissociation occurs at a rate $k_{off}$ in every timestep, regardless of the spatial positions of the particles. The ratio of association and dissociation rate $k_{off}/k_{on}$ is correlated to the equilibrium dissociation constant correlates with $K_d$.

In this model, we set up three types of specific interactions as virtual chemical bonds formed by virtual reactions. As virtual reactions, the model here includes specific interactions between TARPc and PDZ domain, between GluN2Bc and PDZ domain, and between GluN2Bc and CaMKII kinase domain. For simplicity, the model assumes that GluN2Bc binds to the kinase domain near the autoinhibition domain, although it is the autoinhibition domain where GluN2Bc binds to CaMKII in reality. Since the binding sites are different, the specific interaction between GluN2Bc and the PDZ and that between GluN2Bc and the kinase domain of CaMKII are not mutually exclusive; GluN2Bc can simultaneously bind to both the PDZ domain and the kinase domain of CaMKII. However, the specific interaction between TARP and the PDZ domain and that between GluN2Bc and the PDZ domain are mutually exclusive; TARP and GluN2Bc cannot simultaneously bind to the same PDZ domain. In three types of specific interactions, the microscopic rate constant $k_{on}$ in the virtual association reaction is all fixed at 1.0 ns⁻¹ throughout all simulations. In the simulations, trials of whether a reaction occurs or not are performed every timestep $\Delta t$ = 0.25 ns in both the 3D and 2D systems. The microscopic rate constant $k_{off}$ for each virtual dissociation reaction is determined from experimental values of the equilibrium dissociation constant $K_d$ using the same protocol as in the previous study (*Yamada and Takada, 2023*) (see the next section).

Alongside the specific interaction, we introduce a nonspecific interaction $V_{nonspe}$, which applies between any two particles. This interaction combines a weak inter-domain force with an excluded volume effect. The potential function is defined by finite soft-core potentials as follows:

$$V_{\text{nonspe}} = \sum \begin{cases} \frac{1}{2}k(r-d)^2 - E_{\text{nonspe}}, & r < d \\ \frac{E_{\text{nonspe}}}{2}\left(\frac{r_c - d}{2}\right)^{-2}(r-d)^2 - E_{\text{nonspe}}, & d \leq r < d + \frac{r_c - d}{2} \\ -\frac{E_{\text{nonspe}}}{2}\left(\frac{r_c - d}{2}\right)^{-2}(r-r_c)^2, & d + \frac{r_c - d}{2} \leq r < r_c \\ 0, & r \geq r_c \end{cases} \tag{A9}$$

where the potential minimum distance $d$ equals the sum of the particle radii $r_0$, and the cutoff distance ($r_c$) is 1.5 $r_0$. The cutoff distance for nonspecific interactions is here limited up to 1.5 nm. This upper limit of the cutoff 1.5 nm is determined as it best represents the concentration of the dilute phase in the AMPAR(TARP)₄-PSD-95 system in 3D slab simulation. The spring constant is 10 kJ/mol/nm², and the potential depth is adjusted based on experimental data. This nonspecific interaction complements the specific binding model.

## Parameter tuning

The interaction strengths in the mesoscale model are calibrated based on available experimental data in the 3D system; we utilize the equilibrium dissociation constants $K_d$'s and the critical concentration of phase separation reported in in vitro experiments. The tuned parameter set is used in the production runs of both the 3D and 2D systems.

For a pair of AMPAR(TARP)$_4$ and PSD-95, we refer to the experimental $K_d$ for TARPc and PSD-95 being 2.66 μM (*Vistrup-Parry et al., 2021*), and the critical concentrations for the phase separation to be $\lesssim$ 5 μM (*Vistrup-Parry et al., 2021*; *Zeng et al., 2019*). Using the data, we simultaneously tune strengths of specific and non-specific interactions as in our previous work (*Yamada and Takada, 2023*). We modulate the specific interaction strength by changing the off rate $k_{off}$ values in the virtual reaction for binding. For the non-specific interaction, we modulate the distance parameter $r_c$ in the nonspecific pairwise potential. In the 2-dimensional parameter space ($k_{off}$, $r_c$), we repeat a binding-dissociation simulation of a binary system containing one TARPc particle and one PSD-95 molecule, from which we estimate computational $K_d$ values. The computational $K_d$ agrees with the experimental $K_d$ = 2.66 μM along a line in the parameter space. Then, on several points along the line, we repeatedly perform simulations that contain 200 copies of AMPAR(TARP)$_4$ and PSD-95 in the 50 × 50 × 500 nm elongated simulation box. In these simulations, we obtain configurations well separated into the dilute and highly condensed phases and measure the average concentration of PSD-95 in the dilute phase, which is equivalent to the computational critical concentration of the phase separation. We identify the set of parameters ($k_{off}$, $r_c$)=(4.0 × 10$^{-4}$ ns$^{-1}$, 1.5 nm) that result in a good agreement of computational (2.69 μM) and experimental ($\lesssim$ 5 μM) critical concentration. The value of $r_c$ is used for all the particle pairs.

Next, for the remaining two pairs of two-component systems (GluN2Bc-PSD-95 and CaMKII-GluN2Bc), we tune the parameters for specific interactions by changing the virtual reaction rate constant $k_{off}$. The $K_d$ for GluN2Bc and PSD-95 is reported as 1.19 μM (*Vistrup-Parry et al., 2021*), while that for GluN2Bc and the active CaMKII dodecamer is measured to be 0.107 μM (*Özden et al., 2022*). In the same way as above, using a binary system, we calibrate the $k_{off}$ values (*Appendix 1—figure 1*) to be 1.0 × 10$^{-4}$ ns$^{-1}$ for GluN2Bc-PSD-95, and 6.0 × 10$^4$ ns$^{-1}$ for CaMKII-GluN2Bc, respectively. For more detailed calibration methods, including dilute phase measurements and single molecule simulations, refer to the methods in our previous paper (*Yamada and Takada, 2023*).

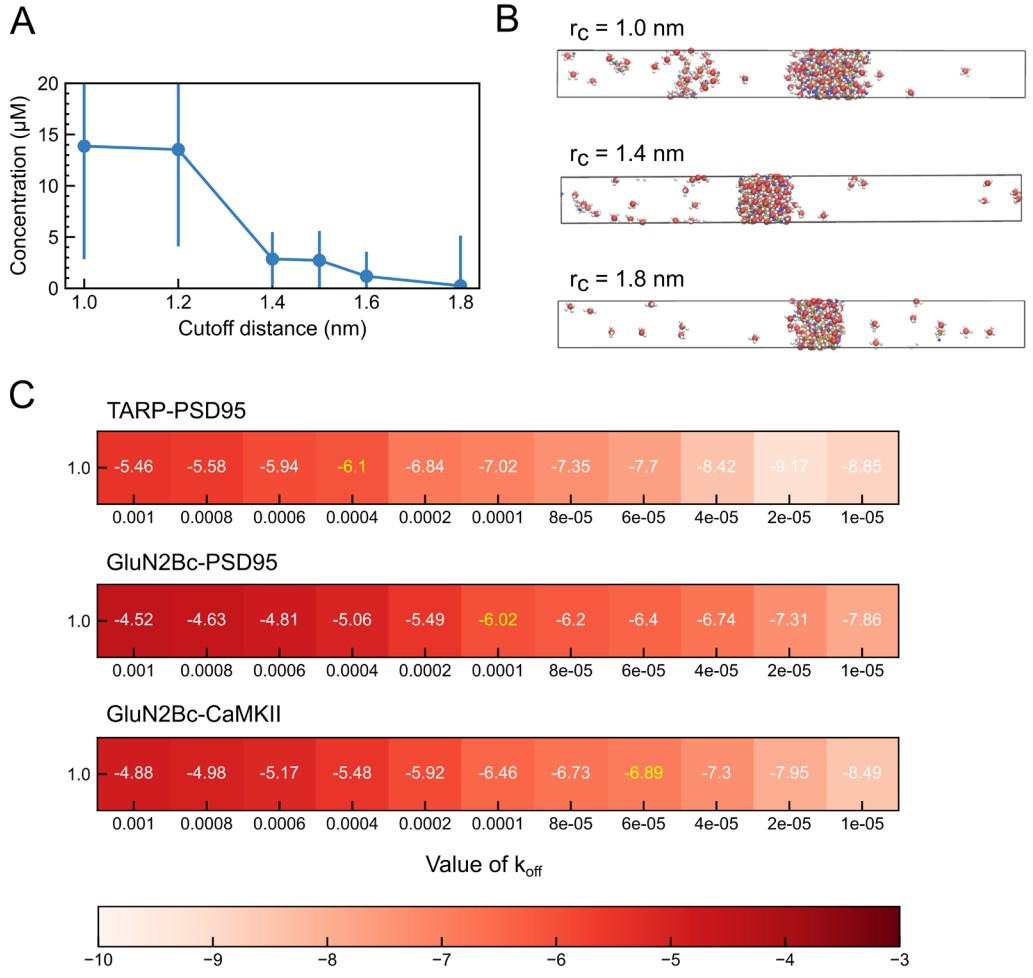

**Appendix 1—figure 1.** Validation results of our mesoscale model. (**A**) Estimation results of the concentration of PSD-95 in the dilute phase for each upper cutoff distance limit. Among the cutoff value from 1.0 nm to 1.8 nm, we choose 1.5 nm as a cutoff limit since it was the shortest distance which almost reproduces the critical concentration of PSD-95. (**B**) Snapshots of the simulation result in three cutoff conditions: $r_c$ = 1.0 nm (top), $r_c$ = 1.4 nm (middle), and $r_c$ = 1.8 nm (bottom). The volume of the condensed phase decreases as the cutoff distance decreases. (**C**) Estimated dissociation constants ($K_d$) between AMPAR-PSD-95 (top), GluN2Bc-PSD-95 (middle), and GluN2Bc-CaMKII kinase (bottom), calculated by using single molecule simulations based on our previous methods (*Yamada and Takada, 2023*). The numbers in the boxes represent the logarithm of the dissociation constant. For the molecular dissociation reaction in each molecular pair, the X-axis value of $k_{off}$, which takes the value of the dissociation constant shown in yellow color in the diagram, was adopted as the reaction rate constant and used in the subsequent simulations.

## Simulation in the 3D system

In the 3D system containing four kinds of protein complexes, as an initial configuration, we place 135 molecules of DsRed2(TARPc)$_4$, 240 molecules of PSD-95, 90 molecules of eqFP670(GluN2Bc)$_2$, and 60 molecules of CaMKII randomly avoiding their mutual overlaps, in a 155.362 × 155.362 × 155.362 nm cubic box with the periodic boundary condition. *Appendix 1—table 1* summarizes the size of the system, the number of each molecule in the system, and its average monomeric concentration. The amount of PSD-95 is determined so that the total amount of PDZ domains is equal to the amount of its binding clients (sum of TARPc and GluN2Bc). This setup theoretically allows the PSD-95 and the two types of receptors to bind without excess or deficiency, while the amount of CaMKII is in excess relative to the possible binding sites of NMDARs. We note that the relative fraction of PSD-95 is lower than that in the in vitro experiment (*Hosokawa et al., 2021*) (We

confirm that excess amount of PSD-95 results in dissolution of excess PSD-95 molecules without changing condensate morphology). To investigate the morphology of the two phases, we adopt a cubic box system, rather than an elongated box; the concentrations of the dilute phases may be subject to artifacts from the finite size and thus not be very accurate as an estimate of critical concentration of LLPS. Then, three independent simulations at 300 K with different stochastic forces are repeated for $8 \times 10^7$ MD timesteps.

**Appendix 1—table 1.** Conditions of concentrations in 3D system.

|  | AMPAR | NMDAR | PSD-95 | CaMKII | $L_{3D}$ |
|---|---|---|---|---|---|
| Copies | 135 | 90 | 240 | 60 | 155.362 nm |
| Monomeric conc. | 239 μM | 79.7 μM | 106 μM | 319 μM |  |
| Conc. in experiments | 7.5 μM | 2.5 μM | 10 μM | 10 μM |  |

With the above setup, we performed the simulations starting from a random configuration as explained in the main text. In the simulations, the average concentration of each molecule in the dilute phase is shown in *Appendix 1—table 2*. The solubility product value obtained from all the dilute phase concentrations of each molecule ($K_{sp}$) is almost the same as the concentration conditions for the monomer given in the studies of Hosokawa and co-workers (*Hosokawa et al., 2021*). It should be noted that the number of molecules except CaMKII in the dilute phase is very small in the simulation system. A difference of one molecule corresponds to the difference in ~1 μM in the system, which is on the same order as the mean values of concentrations. Thus, the estimate inevitably contains large error. With the standard error in mind, the solubility product ($K_{sp}$) in the dilute phase is in good agreement with the experiment.

**Appendix 1—table 2.** Concentrations of each molecule in the dilute phase.

|  | AMPAR | NMDAR | PSD-95 | CaMKII | $K_{sp}$ |
|---|---|---|---|---|---|
| Dilute conc. in simulations | 7.17 μM | 1.92 μM | 1.03 μM | 141 μM | 1999 μM$^4$ |
| Standard error | ±0.62 μM | ±0.39 μM | ±0.32 μM | ±7.9 μM | ±1329 μM$^4$ |
| Conc. in experiments | 7.5 μM | 2.5 μM | 10 μM | 10 μM | 1875 μM$^4$ |

## Simulation in the 2D system

In the 2D system containing four kinds of protein complexes, AMPAR(TARP)$_4$ and NMDAR(GluN2Bc)$_2$ are embedded to the membrane, PSD-95 is tethered to the membrane via the palmitoyl group, and CaMKII is in the cytoplasm beneath the membrane. *Appendix 1—table 3* summarizes the size of the system, the number of each molecules in the 2D simulation system, those corresponding literature average values (*Fukazawa and Shigemoto, 2012*; *Nair et al., 2013*; *Sheng and Hoogenraad, 2007*; *Shinohara, 2012*) per postsynaptic spine. Note that literature values may vary depending on the region of the brain where the synapse is located and the type of organism. In a 500 × 500 × 500 nm cubic box (the periodic boundary condition), we set the plane at the center of the box (Z=0) as a center of the postsynaptic membrane and apply a potential energy for the membrane. Transmembrane proteins AMPARs and NMDARs and the palmitoyl group of PSD-95 are assumed to diffuse only on the membrane due to this membrane potential. Other domains of PSD-95 and CaMKII can move around in the cytoplasm below the membrane (Z < -2.5 nm). Starting from the randomly placed initial conditions, simulations at 300 K are performed for $8 \times 10^7$ MD time steps.

**Appendix 1—table 3.** Conditions of molecular number in 2D system.

|  | AMPAR | NMDAR | PSD-95 | CaMKII | $L_{2D}$ |
|---|---|---|---|---|---|
| Copies | 135 | 90 | 240 | 60 | 500 nm |
| Literature values | ~50–100 (*Fukazawa and Shigemoto, 2012*; *Nair et al., 2013*) | ~20–30 (*Shinohara, 2012*) | 300 (*Sheng and Hoogenraad, 2007*) | 467 (*Sheng and Hoogenraad, 2007*) |  |

In the analysis, we calculate the orientation of CaMKII relative to the membrane plane as follows. First, for each six kinase domains in one side of CaMKII, we define the vector normal to the approximate hexagonal plane by adding the six vectors from the positions of six kinase domains to the central hub domain. Then, we estimate the angle between the hexagonal plane and the membrane plane. The angle is calculated for each six kinases, and thus two angles are obtained per one CaMKII complex. These angles for all the CaMKII molecules have been taken to get the orientation distribution.

## Measurement of interfacial tension in the 3D slab system

To measure the interfacial tensions between separated phases in the 3D system, we perform a set of simulations in the slab configurations. We aim to obtain the interfacial tensions of the three interfaces; the interfacial tension $\gamma_A$ between the dilute phase and the AMPAR/PSD-95 condensate phase (denoted as the 'A' phase), the tension $\gamma_N$ between the dilute phase and the NMDAR/PSD-95/CaMKII condensate phase (the 'N' phase), and the tension $\gamma_{AN}$ between the 'A' and 'N' condensate phases. For the estimate of $\gamma_A$ and $\gamma_N$, we prepare the slab configurations of the A and N phases, respectively, within the dilute phase. For the estimate of $\gamma_{AN}$, we set up a three-phase slab configuration in which the A phase is next to the N phase, and these phases are embedded in the dilute phase (see *Figure 6A*).

For each measurement of $\gamma_A$ and $\gamma_N$, we first conduct the simulation in the $50 \times 50 \times 500$ nm periodic box for $4 \times 10^6$ MD timestep, confining the molecules written in *Appendix 1—table 4* for each condition into a small region of $50 \times 50 \times 100$ nm by applying a confinement potential. To measure the surface tension of each of the multiphases observed in the 3D cubic simulation system, the number of molecules in the slab box is chosen to meet the same stoichiometry of each molecule included in the phase in 3D cubic box. The ratio of molecules, therefore, differs from that of the 3D cubic box system described formerly. The concentration of all types of molecules is also described in *Appendix 1—table 4*. Then, we remove the confinement potential and continue the simulation for $2 \times 10^7$ MD timesteps in an elongated $50 \times 50 \times 1500$ nm box with the periodic boundary condition. From the pressure tensor obtained in the simulations, we estimate interfacial tension value between respective condensed and dilute phases, using Kirkwood-Buff equation (*Ismail et al., 2006*; *Kirkwood and Buff, 1949*) as follows:

$$\gamma = \frac{1}{2} L_z \left\{ P_{zz} - \frac{1}{2} \left( P_{xx} + P_{yy} \right) \right\}$$

(A10)

In *Appendix 1—table 4*, slightly different amount of CaMKII is included for different radius, based on the number of CaMKII molecules contained in the cluster. The amount of CaMKII in each condition is corresponding to the result obtained from simulations in a cubic box. It should be noted that we have confirmed that the value of interfacial tension does not change dramatically even when the CaMKII content was fixed (*Figure 6—figure supplement 1*).

For the measurement of $\gamma_{AN}$, we place 'N' phase in a region of 0 nm $<Z<$ 100 nm and 'A' phase in a region of $-100$ nm $<Z<$ 0 nm, respectively, as an initial configuration. The simulation is conducted for $2 \times 10^7$ MD time steps in a $50 \times 50 \times 1500$ nm box with the periodic boundary condition. The interfacial tension is calculated by subtracting $\gamma_A$ and $\gamma_N$ values obtained in the above simulations.

$$\gamma_{AN} = L_z \left\{ P_{zz} - \frac{1}{2} \left( P_{xx} + P_{yy} \right) \right\} - \left( \gamma_A + \gamma_N \right)$$

(A11)

**Appendix 1—table 4.** Conditions of molecular number in 2D system.

|  | AMPAR | NMDAR | PSD-95 | CaMKII | elongated box length |
|---|---|---|---|---|---|
| $\gamma_A$ | 270 | 0 | 324 | 0 | 1500 nm |
| $\gamma_N$ (R = $1r$) | 0 | 180 | 120 | 80 | 1500 nm |
| $\gamma_N$ (R = $(2/3)\,r$) | 0 | 180 | 120 | 90 | 1500 nm |

*Appendix 1—table 4 Continued on next page*

Appendix 1—table 4 Continued

|  | AMPAR | NMDAR | PSD-95 | CaMKII | elongated box length |
|---|---|---|---|---|---|
| $\gamma_N \ (R = (1/2)\, r)$ | 0 | 180 | 120 | 100 | 1500 nm |
| $\gamma_{AN}$ | 270 | 180 | 480 | 120 | 1500 nm |
| Max Conc. | 120 μM | 79.7 μM | ~212 μM | ~53.1 μM |  |

