## [Editor Report · eLife Assessment]

This **important** study provides a conceptual advance in our understanding of how membrane geometry modulates the balance between specific and non-specific molecular interactions, reversing multiphase morphologies in postsynaptic protein assemblies. Using a mesoscale simulation framework grounded in experimental binding affinities, the authors successfully recapitulate key experimental observations in both solution and membrane-associated systems, providing novel mechanistic insight into how spatial constraints regulate postsynaptic condensate organization. The conclusions are supported by **solid** strength of evidence and the findings are of broad significance for both computational and experimental biologists

---

## [Referee Report · Reviewer #2 (Public review)]

This is a timely and insightful study aiming to explore the general physical principles for the sub-compartmentalization--or lack thereof--in the phase separation processes underlying the assembly of postsynaptic densities (PSDs), especially the markedly different organizations in three-dimensional (3D) droplets on one hand and the two-dimensional (2D) condensates associated with a cellular membrane on the other. Simulation of a highly simplified model (one bead per protein domain) is apparently carefully executed. Based on a thorough consideration of various control cases, the main conclusion regarding the trade-off between repulsive excluded volume interactions and attractive interactions among protein domains in determining the structures of 3D vs 2D model PSD condensates is quite convincing. The novel results in this manuscript should be published.

Comment on the revised manuscript:

The authors have adequately addressed all my previous concerns. The manuscript is now much improved, ready for publication as a version of record.

---

## [Referee Report · Reviewer #3 (Public review)]

Summary:

In this work, Yamada, Brandani and Takada have developed a mesoscopic model of the interacting proteins in the postsynaptic density. They have performed simulations, based on this model and using the software ReaDDy, to study the phase separation in this system in 2D (on the membrane) and 3D (in the bulk). They have carefully investigated the reasons behind different morphologies observed in each case, and have looked at differences in valency, specific/non-specific interactions and interfacial tension.

Strengths:

The simulation model is developed very carefully, with strong reliance on binding valency and geometry, experimentally measured affinities, and physical considerations like the hydrodynamic radii. The presented analyses are also thorough, and great effort has been put into investigating different scenarios that might explain the observed effects.

Weaknesses:

The biggest weakness of the study, in my opinion, has been a lack of more in-depth and quantitative physical insights about phase separation theories. In the revised version, the authors have added text to point the interested reader to the respective theories, and have included a qualitative assessment of their findings in the light of said theories. This better positions their discussion. I still believe the role of entropic effects need more attention, which can be the subject of future studies.

The authors have revised their Introduction and added text to the Discussion, to enrich their view on the attractive and repulsive forces as well as mixing entropy. This version better covers the physics of phase separation.

I appreciate the added discussion about the different diffusive behavior in the membrane in contrast to the bulk (i.e. the Saffman-Delbrück model). This paves the way for future studies, including realistic kinetics of the studied system.

---

## [Author Response]

The following is the authors’ response to the original reviews.

**Reviewer #1 (Public review):**
Summary:This study uses mesoscale simulations to investigate how membrane geometry regulates the multiphase organization of postsynaptic condensates. It reveals that dimensionality shifts the balance between specific and non-specific interactions, thereby reversing domain morphology observed in vitro versus in vivo.Strengths:The model is grounded in experimental binding affinities, reproduces key experimental observations in 3D and 2D contexts, and offers mechanistic insight into how geometry and molecular features drive phase behavior.Weaknesses:The model omits other synaptic components that may influence domain organization and does not extensively explore parameter sensitivity or broader physiological variability.

We thank the reviewer for his/her time and effort to our manuscript. We agree with the point that the contribution of other synaptic components should be addressed. We have included a discussion of the effects of environmental factors such as protein and ion concentrations, as well as other omitted postsynaptic components (SAPAP, Shank, and Homer) on phase morphology. In the middle of the 2^nd^ paragraph of Discussion, we added:

“While these in vivo results contain additional scaffold and cytoskeletal elements omitted in our model, such as SAPAP, Shank and Homer, nearly all proteins in the middle and lower layers of the PSD associate directly or indirectly with PSD-95 in the upper PSD layer. Consequently, it is probable that other scaffold proteins contribute to the mobility of AMPAR-containing and NMDAR-containing nanodomains indistinguishably. They may increase the stability of the AMPAR and NMDAR clusters but are unlikely to have a distinct effect to reverse the phase-separation phenomenon.”

Also, as the reviewer pointed out, we agree with that physiological factors such as ion concentration may influence the phase. However, conditions such as ion concentration are implicitly implemented as the specific and nonspecific interactions in this model, which makes it difficult to estimate the effect of each physiological condition individually. We added the variability potential of physiological conditions to the discussion section as a limitation of this model. To investigate parameter sensitivity in more detail, we performed additional MD simulations with weakened membrane constraints to account for the behavior between 3D and 2D. We added:

“First, our results did not provide direct insights to physiological conditions, such as ion concentrations. Since such factors are implicitly implemented in our model, it is difficult to estimate these effects individually. This suggests the need for future implementation of environmental factors and validation under a broader range of in vivo-like settings.”

**Reviewer #2 (Public review):**
This is a timely and insightful study aiming to explore the general physical principles for the sub-compartmentalization--or lack thereof--in the phase separation processes underlying the assembly of postsynaptic densities (PSDs), especially the markedly different organizations in three-dimensional (3D) droplets on one hand and the twodimensional (2D) condensates associated with a cellular membrane on the other. Simulation of a highly simplified model (one bead per protein domain) is carefully executed. Based on a thorough consideration of various control cases, the main conclusion regarding the trade-off between repulsive excluded volume interactions and attractive interactions among protein domains in determining the structures of 3D vs 2D model PSD condensates is quite convincing. The results in this manuscript are novel; however, as it stands, there is substantial room for improvement in the presentation of the background and the findings of this work. In particular,(i) conceptual connections with prior works should be better discussed(ii) essential details of the model should be clarified, and(iii) the generality and limitations of the authors' approach should be better delineated.

We appreciate the reviewer for his/her time and effort on our manuscript and for encouraging comments and helpful suggestions. We answered every technical comment the reviewer mentioned below.

Specifically, the following items should be addressed (with the additional references mentioned below cited and discussed):(1) Excluded volume effects are referred to throughout the text by various terms and descriptions such as "repulsive force according to the volume" (e.g., in the Introduction), "nonspecific volume interaction", and "volume effects" in this manuscript. This is somewhat curious and not conducive to clarity, because these terms have alternate or connotations of alternate meanings (e.g., in biomolecular modeling, repulsive interactions usually refer to those with longer spatial ranges, such as that between like charges). It will be much clearer if the authors simply refer to excluded volume interactions as excluded volume interactions (or effects).

Thank you for this comment. We have substituted the words “excluded volume interactions” for words of similar meaning. However, we have left the expression of “non-specific interactions” as they are referring to explicit interactions that are given as force fields in the model, rather than in the general meaning of excluded volume effect.

(2) In as much as the impact of excluded volume effects on subcompartmentalization of condensates ("multiple phases" in the authors' terminology), it has been demonstrated by both coarse-grained molecular dynamics and field-theoretic simulations that excluded volume is conducive to demixing of molecular species in condensates [Pal et al., Phys Rev E 103:042406 (2021); see especially Figures 4-5 of this reference]. This prior work bears directly on the authors' observation. Its relationship with the present work should be discussed.

We appreciate the reviewer’s insightful comment. We have now included a more detailed discussion on excluded volume effect in the revised manuscript, which provides important context for our findings. Furthermore, we have cited the references to support and enrich the discussion, as recommended.

(3) In the present model setup, activation of the CaMKII kinase affects only its binding to GluN2Bc. This approach is reasonable and leads to model predictions that are essentially consistent with the experiment. More broadly, however, do the authors expect activation of the CaMKII kinase to lead to phosphorylation of some of the molecular species involved with PSDs? This may be of interest since biomolecular condensates are known to be modulated by phosphorylation [Kim et al., Science 365:825-829 (2019); Lin et al, eLife 13:RP100284 (2025)].

We agree that phosphorylation effect on phase separation is an important and interesting aspect to consider. Some experimental results have shown that activation of CaMKII can lead to phosphorylation of various proteins and make PSD condensate more stable by altering their interactions. We included the sentence below in limitations:

“In this context, we also do not explicitly account for downstream phosphorylation events. Although such proteins are not included in the current components, they will regulate PSD-95, affecting its binding valency, or diffusion coefficient. This is a subject worthy of future research.”

(4) The forcefield for confinement of AMPAR/TARP and NMDAR/GluN2Bc to 2D should be specified in the main text. Have the authors explored the sensitivity of their 2D findings on the strength of this confinement?

We thank the reviewer for the helpful recommendation. We have revised the manuscript to include membrane-mimicking potential on main text. Furthermore, we also think that exploring the shape of the 3D/2D condensate phase due to the sensitivity of confinement is a very interesting point. We have additionally performed MD simulations with smaller/larger membrane constraints and included the results in supporting information as Figure S5. The following parts are added:

“We further attempted to mimic intermediate conditions between 3D and 2D systems in two different manners. First, we applied a weaker membrane constraint in 2D system. Even when the strength of membrane constraints is reduced by a factor of 1000, NMDARs are located on the inner side when the CaMKII was active, as well as the result in 2D system (Fig.S5ABC). Second, to weaken further the effect of membrane constraints, we artificially altered the membrane thickness from 5 nm to 50 nm, in addition to reducing the membrane constraints by 1000. As a result, NMDAR clusters move to the bottom and surround AMPAR (Fig.S5DEF). In this artificial intermediate condition, both states in which the NMDARs are outside (corresponding to 3D) and in which the NMDARs are inside (corresponding to 2D) are observed, depending on the strength of the membrane constraint.”

(5) Some of the labels in Figure 1 are confusing. In Figure 1A, the structure labeled as AMPAR has the same shape as the structure labeled as TARP in Figure 1B, but TARP is labeled as one of the smaller structures (like small legs) in the lower part of AMPAR in Figure 1A. Does the TARP in Figure 1B correspond to the small structures in the lower part of AMPAR? If so, this should be specified (and better indicated graphically), and in that case, it would be better not to use the same structural drawing for the overall structure and a substructure. The same issue is seen for NMDAR in Figure 1A and GluN2Bc in Figure 1B.(6) In addition to clarifying Figure 1, the authors should clarify the usage of AMPAR vs TARP and NMDAR vs GluN2Bc in other parts of the text as well.(7) The physics of the authors' model will be much clearer if they provide an easily accessible graphical description of the relative interaction strengths between different domain-representing spheres (beads) in their model. For this purpose, a representation similar to that given by Feric et al., Cell 165:1686-1697 (2016) (especially Figure 6B in this reference) of the pairwise interactions among the beads in the authors' model should be provided as an additional main-text figure. Different interaction schemes corresponding to inactive and activated CAMKII should be given. In this way, the general principles (beyond the PSD system) governing 3D vs 2D multiple-component condensate organization can be made much more apparent. \

We sincerely appreciate the reviewer’s comments. According to the recommendation, we have changed the diagram in Figure 1B into interaction matrix with each mesoscale molecular representation and the expression in main text to be clearer about AMPAR and TARP, and about the relationship between NMDAR and GluN2Bc. Former diagram of the pairs of specific interaction is moved to supplementary figure.

(8) Can the authors' rationalization of the observed difference between 3D and 2D model PSD condensates be captured by an intuitive appreciation of the restriction on favorable interactions by steric hindrance and the reduction in interaction cooperativity in 2D vs 3D?

We thank the reviewer for the comment. As pointed out, the multiphase morphology change observed in this study can be attributed to a decrease in coordination number in 2D compared to 3D. We have included the physicochemical rationalization in the discussion.

(9) In the authors' model, the propensity to form 2D condensates is quite weak. Is this prediction consistent with the experiment? Real PSDs do form 2D condensates around synapses.

We are grateful to the reviewer for highlighting this important point. We agree with that the real PSD forms 3D condensates beneath the 2D membrane. Some lower PSD components under the membrane (i.e. SAPAP, Shank, and Homer) are omitted in our system, which may cause a weak condensation. To emphasize this, we have added the following sentence:

“While these in vivo results contain additional scaffold and cytoskeletal elements omitted in our model, such as SAPAP, Shank and Homer, nearly all proteins in the middle and lower layers of the PSD associate directly or indirectly with PSD-95 in the upper PSD layer. Consequently, it is probable that other scaffold proteins contribute to the mobility of AMPAR-containing and NMDAR-containing nanodomains indistinguishably. They may increase the stability of the AMPAR and NMDAR clusters but are unlikely to have a distinct effect to reverse the phase-separation phenomenon.”

However, we believe that the clusters formed on the 2D membrane are not a robust “phase” because they do not follow scaling law. In fact, in our previous study of PSD system with AMPAR(TARP)_4_ and PSD-95, we have already reported that phase separation is less likely to occur in 2D than in 3D. The previous result suggests that phase separation on membrane may be difficult to achieve, which is consistent with the results of this study.

(10) More theoretical context should be provided in the Introduction and/or Discussion by drawing connections to pertinent prior works on physical determinants of co-mixing and de-mixing in multiple-component condensates (e.g., amino acid sequence), such as Lin et al., New J Phys 19:115003 (2017) and Lin et al., Biochemistry 57:2499-2508 (2018).(11) In the discussion of the physiological/neurological significance of PSD in the Introduction and/or Discussion, for general interest it is useful to point to a recently studied possible connection between the hydrostatic pressure-induced dissolution of model PSD and high-pressure neurological syndrome [Lin et al., Chem Eur J 26:11024-11031 (2020)].

We thank the reviewer for the helpful recommendation. We have added the recommended references in each relevant part in introduction, respectively.

(12) It is more accurate to use "perpendicular to the membrane" rather than "vertical" in the caption for Figure 3E and other such descriptions of the orientation of the CaMKII hexagonal plane in the text.

We thank you for your comment. We replaced the word “vertical” with “perpendicular" in the main text and caption.

**Reviewer #3 (Public review):**
Summary:In this work, Yamada, Brandani, and Takada have developed a mesoscopic model of the interacting proteins in the postsynaptic density. They have performed simulations, based on this model and using the software ReaDDy, to study the phase separation in this system in 2D (on the membrane) and 3D (in the bulk). They have carefully investigated the reasons behind different morphologies observed in each case, and have looked at differences in valency, specific/non-specific interactions, and interfacial tension.Strengths:The simulation model is developed very carefully, with strong reliance on binding valency and geometry, experimentally measured affinities, and physical considerations like the hydrodynamic radii. The presented analyses are also thorough, and great effort has been put into investigating different scenarios that might explain the observed effects.Weaknesses:The biggest weakness of the study, in my opinion, has to do with a lack of more in-depth physical insight about phase separation. For example, the authors express surprise about similar interactions between components resulting in different phase separation in 2D and 3D. This is not surprising at all, as in 3D, higher coordination numbers and more available volume translate to lower free energy, which easily explains phase separation. The role of entropy is also significantly missing from the analyses. When interaction strengths are small, entropic effects play major roles. In the introduction, the authors present an oversimplified view of associative and segregative phase transitions based on the attractive and repulsive interactions, and I'm afraid that this view, in which all the observed morphologies should have clear pairwise enthalpic explanations, diffuses throughout the analysis. Meanwhile, I believe the authors correctly identify some relevant effects, where they consider specific/nonspecific interactions, or when they investigate the reduced valency of CaMKII in the 2D system.

We thank the reviewer for the insightful and constructive comments. Regarding the difference in phase behavior between 2D and 3D systems, we appreciate the reviewer’s clarification that differences in coordination number and entropy in higher dimensions can account for the observed morphology of the phases. While it may be clear that entropy decreases due to the decrease of coordination number, our objective was to uncover how such an isotropic entropy reduction regulates the behavior of each phase driven by different interactions, which remains largely unknown. To emphasize this, we modified the introduction and have now included a discussion of the entropic contributions to phase behavior in both 2D and 3D systems, and we have made this clearer in the revised manuscript by referencing relevant theoretical frameworks. In the Discussion, we added the sentence below:

“Generally, phase separation can be explained by the Flory-Huggins theory and its extensions: phase separation can be favored by the difference in the effective pairwise interactions in the same phase compared to those across different phases, and is disfavored by mixing entropy. The effective interactions contain various molecular interactions, including direct van der Waals and electrostatic interactions, hydrophobic interactions, and purely entropic macromolecular excluded volume interactions. For the latter, Asakura-Oosawa depletion force can drive the phase separation. Furthermore, the demixing effect was explicitly demonstrated in previous simulations and field theory (61). Importantly, we note that the effective pairwise interactions scale with the coordination number z. The coordination number is a clear and major difference between 3D and 2D systems. In 3D systems, large z allows both relatively strong few specific interactions and many weak non-specific interactions. While a single specific interaction is, by definition, stronger than a single non-specific interaction, contribution of the latter can have strong impact due to its large number. On the other hand, a smaller z in the membrane-bound 2D system limits the number of interactions. In case of limited competitive binding, specific interactions tend to be prioritized compared to non-specific ones. In fact, Fig. 3A clearly shows that number of specific interactions in 2D is similar to that in 3D, while that of non-specific interactions is dramatically reduced in 2D. In the current PSD system, CaMKII is characterized by large valency and large volume. In the 3D solution system, non-specific excluded volume interactions drive CaMKII to the outer phase, while this effect is largely reduced in 2D, resulting in the reversed multiphase.

Also, I sense some haste in comparing the findings with experimental observations. For example, the authors mention that "For the current four component PSD system, the product of concentrations of each molecule in the dilute phase is in good agreement with that of the experimental concentrations (Table S2)." But the data used here is the dilute phase, which is the remnant of a system prepared at very high concentrations and allowed to phase separate. The errors reported in Table S2 already cast doubt on this comparison.

We thank the reviewer for the insightful comment. In the validation process, we adjusted the parameters so that the number of molecules in dilute phase is consistent with the experimental lower limit of phase separation, based on the assumption that phase-separated dilute phase is the same concentration as the critical concentration. That is why we focus on comparing dilute phase concentration in Table S2. However, in our simulations, the number of protein molecules is relatively small since it is based on the average number per synapse spine. For example, there are only about 60 CaMKII molecules at most, and its presence in the dilute phase is highly sensitive to concentration, as the reviewer pointed out. This is one of the limitations, so we have added a description to the Limitations section. We added:

“Second, parameter calibration contains some uncertainty. Previous in vitro study results used for parameter validation are at relatively high concentrations for phase separation, which may shift critical thresholds compared to that in in vivo environments. Also, since the number of molecules included in the model is small, the difference of a single molecule could result in a large error during this validation process.”

Or while the 2D system is prepared via confining the particles to the vicinity of the membrane, the different diffusive behavior in the membrane, in contrast to the bulk (i.e., the Saffman-Delbrück model), is not considered. This would thus make it difficult to interpret the results of a coupled 2D/3D system and compare them to the actual system.

We appreciate the reviewer’s helpful comment. We agree with that there is a concern that the Einstein-Stokes equation does not adequately reproduce the diffusion of membrane-embedded particles. We recalculated the diffusion coefficients for every membrane particle used in this model using the Saffman-Delbrück model and found that diffusion coefficients for receptor cores (AMPAR and NMDAR) were approximately three times larger. These values are still about ~10 times smaller than that of molecules diffusing under the cytoplasm. Additionally, since this study focuses on the morphology of the phase/cluster at the thermodynamic equilibrium, we think that the magnitude of the diffusion coefficient has little influence on the final structure of the cluster. However, we will incorporate the membrane-embedded diffusion as a future improvement item for better modelling and implementation. We added:

“Third, we estimated all the diffusion coefficients from the Einstein-Stokes equation, which may oversimplify membrane-associated dynamics. Applying the Saffmann-Delbrück model to membrane-embedded particles would be desired although the resulting diffusion coefficients remain of the same order of magnitude. These limitations highlight the need for further research, yet they do not undermine the core significance of the present findings in advancing our understanding of multiphase morphologies.”